# A Study of Necessity & Sufficiency of Linear Transformations in the Attention Mechanism

## Abstract

Scaled Dot Product Attention (SDPA) is the backbone of many modern deep-learning models. It is so versatile that it has been used in natural language, vision, and multi-modal domains with very little change compared to its original formulation. This paper studies the linear transformations used in SDPA. To this end, we introduce three variants of the attention mechanism by removing consecutive linear transformations or adding an extra one. We name these variants *Optimized* ($W^V$ removed), *Efficient* ($W^V$ and $W^K$ removed), and *Super Attention* ($W^V$ and $W^K$ removed and $W^A$ introduced) to simplify comparison when referring to them. In addition to providing the mathematical intuition behind these choices, we evaluate these variants when used in the self-attention module of Transformer models on several datasets of varying size and complexity in vision and text modalities for predictive and generative tasks. Optimized and Efficient variants have one and two matrix multiplications fewer per head, respectively, and 25% and 50% fewer parameters, respectively, than standard SDPA. However, the performance change compared to the difference in parameter count is small. Super Attention introduces a new linear transformation on the values, transforming them from the left. It outperforms standard SPDA in both modalities by up to 10% while having one fewer matrix multiplication per head and 25% fewer parameters than standard SPDA. Consequently, it is also faster than standard SDPA.

## 1 Introduction

Not many ideas have had as profound an effect on the field of *Artificial Intelligence* (*AI*) as the *attention mechanism* (Bahdanau et al., 2015). Introduced as a method to improve machine translation, the attention mechanism revolutionized the way neural networks process and interpret data. By allowing models to focus on specific parts of the input while disregarding irrelevant information, it mimics a form of cognitive attention in humans. It not only enhanced the capability and efficiency of Language Models (LM) but also paved the way for the development of advanced AI architectures like the Transformer model (Vaswani et al., 2017).

These advances have had far-reaching impacts, extending beyond Natural Language Processing (NLP) to other areas such as image recognition (Dosovitskiy et al., 2021), autonomous systems (Mott et al., 2019), healthcare (Choi et al., 2016), and multi-modal application Xu et al. (2023).

The formulation of SDPA in all these domains has undergone very little change compared to the original formulation of Vaswani et al. (2017). Instead, "The bigger the better" has been the prevailing maxim in AI in the last few years. Larger Language Models (LLM), such as Llama 3 (Touvron et al., 2023a;b), GPT-4 (Achiam et al., 2023), and Gemini (Anil et al., 2023) have demonstrated unprecedented capabilities in multi-modal domains.

The behemothic sizes of these models have introduced numerous challenges, such as expensive and slow training and inference, leading to secondary problems such as high carbon emission (Dhar, 2020). Furthermore, such models are impossible not only to run but even to store on edge devices such as smartphones, consumer laptops, and even powerful personal workstations.

In recent years, there have been numerous attempts to address this problem using post-training techniques, like quantization (Jacob et al., 2018), Low-Rank Adaptation (LoRA) (Hu et al., 2022), Quantized LoRA (QLoRA) (Dettmers et al., 2023), and sparsification (Ashkboos et al., 2024). There have

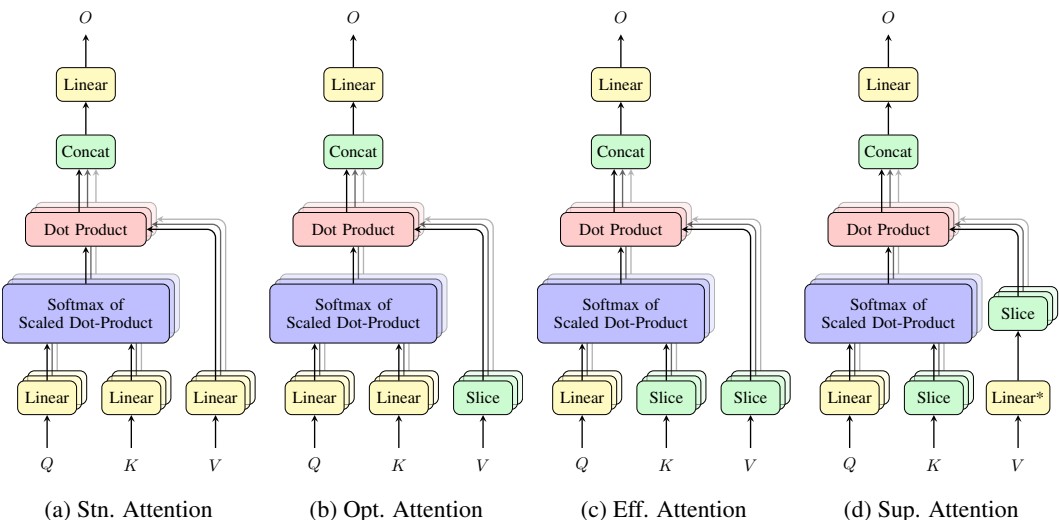

(a) Stn. Attention     (b) Opt. Attention     (c) Eff. Attention     (d) Sup. Attention

Figure 1: Standard multi-head scaled dot product attention (1a) alongside the proposed variations: Optimized Attention (1b), Efficient Attention (1c), and Super Attention (1d). The "Linear" block denotes a linear transformation right while "Linear*" denotes a linear transformation from left.

been also attempts to optimise the speed and GPU utilization of attention-based models. Notable examples include Flash Attention 1, 2, and 3 (Dao et al., 2022; Dao, 2024; Shah et al., 2024).

All these approaches focus on techniques to improve the performance of attention-based models without altering the attention mechanism. In this paper, we look into the attention mechanism itself and study SDPA and three SDPA variants, that are designed based on two intuitive principles: **(1)** two consecutive linear transformations do not introduce non-linearity, and **(2)** a learnable linear kernel between each two inputs of SDPA enhances learning. These three variants are as follows:

◇ **Optimized Attention**, which we introduce in Section 3.1. As shown in Figure 1b, Optimized Attention replaces $W^V$ linear transformation by a simple slicing operation (following Principle 1), thus reducing the number of parameters in the attention layer by 25% and its computational cost by $h$ matrix multiplications, where $h$ is the number of heads. The evaluations in Section 4, show that Optimized Attention reduces the inference time by 2.5–7.5%, while performing similarly (i.e., no/little performance degradation depending on the task).

◇ **Efficient Attention**, which we introduce in Section 3.2. As shown in Figure 1c, Efficient Attention replaces $W^V$ and $W^K$ linear transformations by simple slicing operations (following Principle 1), thus reducing the number of parameters in the attention layer by 50% and its computational cost by $2h$ matrix multiplications, where $h$ is the number of heads. The evaluations in Section 4, show that Efficient Attention reduces the inference time by 5–15%, while performing similarly (i.e., no/little performance degradation depending on the task).

◇ **Super Attention**, which we introduce in Section 3.3. As shown in Figure 1d, Super Attention, introduces a new linear operation $W^A$ (following Principle 2), which transforms the values $V$ from the left. For the sake of simplicity, we build Super Attention on top of Efficient Attention (i.e., $W^V$ and $W^K$ linear transformations are replaced by slicing), but we emphasise that Super Attention can be used on top of standard or Optimized attentions (i.e., without replacing $W^V$ and $W^K$). Super Attention reduces the attention layer's size by $\sim 25\%$ (depending on the attention's context length) and its computational cost by $h$ matrix multiplications. The evaluations in Section 4, show that Super Attention, outperforms standard attention by 2–10% in both vision and NLP tasks (in terms of various learning metrics), while reducing the training and inference time by 2.5–10%.

We evaluate SDPA and our proposed variations in the *self-attention* setting in transformers on **(1)** *image classification* on MNIST, CIFAR100, and ImageNet datasets, **(2)** *natural language sentiment classification* on IMDB and Amazon Reviews datasets, **(3)** *Neural Machine Translation* (*NMT*) on the combined Europarl and Anki English-to-Spanish translation dataset, and **(4)** *generative language modelling* using Andrea Karpathy's NanoGPT on the OpenWebText dataset.

## 2 PRELIMINARIES

Let us start by introducing the notations used throughout the paper. For natural numbers $d_m, d_k \in \mathbb{N}$, we denote the $d_m$-dimensional real *vectors space* by $\mathbb{R}^{d_m}$ and the set of all real $d_m \times d_k$ *matrices* by $\mathbb{R}^{d_m \times d_k}$, noting that all matrices can be regarded as 2D *tensors* and vice versa. Given a set $\mathcal{A} \subseteq \mathbb{R}^{d_m}$, we denote the smallest real vector space containing $\mathcal{A}$ by $\mathrm{span}(\mathcal{A})$. Similarly, given a matrix $W \in \mathbb{R}^{d_m \times d_k}$, we denote the smallest real vector space containing the columns of $W$'s by $\mathrm{span}(W)$. For a *subspace* $\mathcal{S} \leq \mathbb{R}^{d_m}$, the *dimension* of $\mathcal{S}$, denoted $\dim(\mathcal{S})$, is the size of the largest *linearly independent* set in $\mathcal{S}$. The *rank* of a matrix $W \in \mathbb{R}^{d_m \times d_k}$, denoted $\mathrm{rank}(W)$, is the number of linearly independent columns (or rows) in $W$. The rank-nullity theorem implies that $\mathrm{rank}(W) = \dim(\mathrm{span}(W))$ and $\mathrm{rank}(W) \leq \min(d_m, d_k)$.[1]

We use the definition of the attention mechanism as implemented in SotA open-source models, such as Llama-3 and Mistral as well as machine learning frameworks like Torch, JAX, TensorFlow, and Keras. For consistency, we use the same notation as (Vaswani et al., 2017).

**Definition 1** (Standard Attention). The (*multi-head*) *scaled dot-product attention* on *input* tensors $Q, K, V \in \mathbb{R}^{\ell \times d_m}$ is defined as

$$
\begin{aligned}
O &= (H_1\ H_2\ \cdots\ H_h)W^O, &(1)\\
H_i &= S_i V_i', &(2)\\
S_i &= \mathrm{softmax}\left(\frac{Q_i' K_i'^{\mathsf{T}}}{\sqrt{d_k}}\right), &(3)\\
V_i' &= V W_i^V, &(4)\\
K_i' &= K W_i^K, &(5)\\
Q_i' &= Q W_i^Q, &(6)
\end{aligned}
$$

where $O$ is the *output*; $Q_i', K_i', V_i', S_i$, and $H_i$ are the *query*, *key*, *value*, *attention score*, and *head value* of the $i$-th *head*, respectively. The natural numbers $\ell, d_m$ and $h$ are the *context length*, *model dimension*, and *number of heads*, respectively. Moreover, $W_i^Q, W_i^K \in \mathbb{R}^{d_m \times d_k}$ and $W_i^V \in \mathbb{R}^{d_m \times d_v}$, where $d_k$ and $d_v$ are the *key* and *value dimensions*, respectively.

Parameters $d_m, d_k, d_v$ and $h$ are often chosen so that $d_k = d_v = d_m/h$, and in recent models, including SotA Transformer models, $Q, K$, and $V$ are set to $X$, a single input tensor; whereby, the attention mechanism is called *self-attention*.

## 3 REVISING THE ATTENTION MECHANISM

In this section, we discuss our motivation for revisiting the attention mechanism and considering the proposed variants. It is important to note these variants are not mathematically equivalent to standard attention, and our goal here is to justify the choices of variants discussed in this paper. These variants are *Optimized Attention*, *Efficient Attention*, and *Super Attention*, which we introduce in Sections 3.1, 3.2, and 3.3, respectively.

### 3.1 OPTIMIZED ATTENTION: ABSORBING $W_i^V$'S INTO $W^0$

Our objective is to reduce the computational cost and number of parameters in SDPA. Here we focus on (1) and (4) in standard attention. We propose absorbing $W_1^V, W_2^V, \ldots, W_h^V$ into $W^O$, which in turn, reduces the computational cost of the attention layer by $h$ matrix multiplications. But the question is how does this affect the performance of the model. We answer this question in Section 4. Before doing so, however, let us justify our motivation.

---

[1] For a detailed introduction to these see (Meyer, 2023, Chapters 2 & 4).

In standard attention, the output $O$ of the attention layer can be written as

$$O = (H_1 \ H_2 \ \cdots \ H_h)W^O = (S_1 V W_1^V \ S_2 V W_2^V \ \cdots \ S_h V W_h^V) \begin{pmatrix} W_1^O \\ W_2^O \\ \vdots \\ W_h^O \end{pmatrix} \quad (7)$$

$$= S_1 V W_1^V W_1^O + S_2 V W_2^V W_2^O + \cdots + S_h V W_h^V W_h^O,$$

where $W_i^O$ is the matrix that contains rows $(i-1)d_v + 1, \ldots, id_v$ of $W^O$ for $i = 1, 2, \ldots, h$. By the rank-nullity theorem, for each head, we have that

$$\dim(\text{span}(V W_i^V W_i^O)) = \text{rank}(V W_i^V W_i^O) \le \text{rank}(W_i^V W_i^O),$$
$$\le \min(\text{rank}(W_i^V), \text{rank}(W_i^O)) = \min(d_m, d_v) = d_v.$$

In other words, $V W_i^V W_i^O$ has at most $d_v$ independent columns, and the linear function $V \mapsto V W_i^V W_i^O$ maps the columns of $V$ into a $d_v$-dimensional subspace of $\mathbb{R}^{d_m}$.

Thus, standard attention uses two consecutive matrix multiplications to embed the columns of $V$ into a $d_v$-dimensional subspace of $\mathbb{R}^{d_m}$, which goes against Principle 1. Optimized Attention, instead of using two consecutive linear transformations (one downscaling and one upscaling), uses one slicing and one linear transformation as shown in Figure 1b and described in Definition 2.

In more detail, instead of multiplying $V$ from the right by $W_i^V$, we first slice $V$ into $V_1, \ldots, V_h$, where $V_i$ consists of columns $(i-1)d_v + 1, \ldots, id_v$ of $V$, and then, instead of computing $S_i V W_i^V W_i^O$, we compute $S_i V_i W_i^O$, which requires fewer parameters and matrix multiplications (see Remark 1). We have provided a detailed discussion on the computational gains of Optimized Attention in Section 4.3.

**Definition 2** (Optimized Attention). Using the notation of Definition 1, *Optimized Attention* is the attention mechanism defined by the following set of equations:

$$O = (H_1, H_2, \ldots, H_h)W^O, \quad (8)$$
$$H_i = S_i V_i, \quad (9)$$
$$S_i = \text{softmax}(\frac{Q_i' {K_i'}^\mathsf{T}}{\sqrt{d_k}}), \quad (10)$$
$$K_i' = K W_i^K, \quad (11)$$
$$Q_i' = Q W_i^Q. \quad (12)$$

*Remark* 1. Optimized Attention is more efficient than standard attention in the sense that it has $h$ matrix multiplication and $d_m^2$ parameters fewer than standard attention.

*Proof.* Compared to Optimized Attention, standard attention has extra $W_1^V, W_2^V, \ldots, W_h^V$, which are multiplied from the right to $V$, amounting to a total of $d_m d_v h = d_m^2$ parameters and $h$ matrix multiplications. $\square$

## 3.2 EFFICIENT ATTENTION: ABSORBING $W^K$ INTO $W^Q$

In the last section, we discussed our motivation behind removing $W^V$. Here, we repeat the same thing for $W^K$ to further reduce the computational cost of the attention mechanism. When computing the pre-softmax attention scores for each head, we have that

$$\dim(\text{span}(\frac{Q W_i^Q {W_i^K}^\mathsf{T} K^\mathsf{T}}{d_k})) = \text{rank}(Q W_i^Q {W_i^K}^\mathsf{T} K^\mathsf{T}) \le \text{rank}(W_i^Q {W_i^K}^\mathsf{T}),$$
$$\le \min(\text{rank}(W_i^Q), \text{rank}(W_i^K)) = \min(d_m, d_k) = d_k.$$

More precisely, here two linear kernels $W_i^Q$ and $W^{K\intercal}_i$ are stacked, which goes against Principle 1. Thus, in a similar fashion to what we did in Optimized Attention, we merge $W^{K\intercal}_i$ into $W_i^Q$ by replacing the $W_i^K$ linear transformation by slicing as depicted in Figure 1c and defined in Definition 3.

**Definition 3** (Efficient Attention). Using the same notation as Definition 2, *Efficient Attention* is defined via the following equations:

$$
\begin{aligned}
O &= (H_1, H_2, \ldots, H_h)W^O, & (13) \\
H_i &= S_i V_i, & (14) \\
S_i &= \mathrm{softmax}(\frac{Q_i' K_i^\intercal}{\sqrt{d_k}}), & (15) \\
Q_i' &= Q W_i^Q, & (16)
\end{aligned}
$$

where $K_i$ denotes the subtensor consisting of $(i-1)d_k + 1, \ldots, id_k$ rows from $K$.

*Remark* 2. Efficient Attention is more efficient than Optimized Attention and standard attention in the sense that it has $h$ matrix multiplication and $d_m^2$ parameters fewer than Optimized Attention and $2h$ multiplication and $2d_m^2$ parameters fewer than standard attention.

*Proof.* In Efficient Attention, we do not have $W_1^K, W_2^K, \ldots, W_h^K$, which are applied to $K$ from left. Hence, compared to Optimized Attention, we have reduced the number of matrix multiplications by $h$ and parameters by $d_m^2$. From this and Remark 1, it follows that Efficient Attention has $h + h = 2h$ matrix multiplication and $d_m^2 + d_m^2 = 2d_m^2$ parameters less than standard attention. □

### 3.3 SUPER ATTENTION: INTRODUCING $W^A$

Looking at the Equations (1-6), we observe that in SDPA, there are learnable parameters between $Q$ and $K$; however, there is no such parameter between $K$ and $V$ (even though a $\mathrm{softmax}$ is applied to the term containing $K$). Thus, we introduce a new learnable parameter $W^A$ which linearly transforms the values from left. To better observe this, let us write the equation for one head in one of the attention variants, e.g., Efficient Attention by combining Equations (14–16):

$$
H_i = \mathrm{softmax}(\frac{Q W_i^Q K_i^\intercal}{d_m}) V_i W^O. \tag{17}
$$

As we see in Equation (17), there are no learnable parameters between $K^\intercal$ and $V$, and the attention scores $S_i$ are directly applied to the values $V_i$. The intuition behind directly applying $S_i$ to $V_i$ is that the attention scores in $S_i$ determine "how much attention is paid" to each of the features of each token in $V_i$. Despite this intuition, we found that in practice the model can benefit from an additional kernel which appears in between the scores $S_i$ and values $V_i$. Specifically, with the introduction of $W^A$, Equation (17) changes to

$$
H_i = \mathrm{softmax}(\frac{Q W_i^Q K_i^\intercal}{d_m}) W^A V_i W^O. \tag{18}
$$

The role of $W^A$ is to mix and align the values vertically (token-wise). Thus, to prevent "look ahead" in the attention mechanism for use in generative language modelling, we constrain $W^A$ to be lower triangular, so that future tokens do not influence the current one in $W^A$. Note that we use the same $W^A$ for all heads. The reason here is that we want to improve the model performance while keeping the model size as small as possible. Thus, in a more general formulation, one can use different $W^A$ for each head to perhaps gain better performance, but at the cost of increasing the number of parameters, and thereby the model size.

**Definition 4** (Super Attention). Using the notation of Definition 3, *Super Attention* is the attention mechanism defined by the following set of equations:

$$
\begin{align}
O &= (H_1, H_2, \ldots, H_h)W^O, \quad &(19) \\
H_i &= S_i V_i', \quad &(20) \\
S_i &= \mathrm{softmax}\left(\frac{Q_i' K_i^\intercal}{\sqrt{d_k}}\right), \quad &(21) \\
V_i' &= W^A V_i, \quad &(22) \\
Q_i' &= Q W_i^Q, \quad &(23)
\end{align}
$$

where $W^A \in \mathbb{R}^{\ell \times \ell}$ is the *alignment kernel*, which vertically (i.e., for values corresponding to different tokens) aligns and mixes the values before the attention scores are applied to them.

*Remark* 3. Super Attention is more efficient than standard attention whenever the model dimension $d_m$ is greater than or equal to the context length $\ell$. This means that Super Attention has at least $h$ matrix multiplication and $d_m^2$ parameters fewer than standard attention.

*Proof.* Looking at the Equations (13–16) and (19–23), we observe that Super Attention and Efficient Attention have the same defining equations, except that Super Attention has an additional linear transformation in Equation (22), where $V_i$'s are multiplied by $W^A$ from the left. This amounts to $\ell^2$ parameters and $h$ matrix multiplication more than Efficient Attention. From Remark 2, it follows that Super Attention has at least $2d_m^2 - \ell^2 \geq d_m^2$ parameters and $2h - h = h$ matrix multiplications less than standard attention. $\qquad\square$

## 4 Evaluation

We evaluate all the proposed mechanisms in vision (Section 4.1) and NLP (Section 4.2 and Appendix A.5). We also provided a detailed comparison of the computational costs and edge device performance in Section 4.3 and Appendices A.1 and A.2.

**Evaluation Methodology.** We have chosen various benchmarks to ensure a fair and comprehensive comparison between the four attention mechanisms discussed. In each benchmark, we have followed the common practices used to evaluate the performances. For all benchmark, (1) we use the same model architecture and iterate between standard, Optimized, Efficient, and Super Attention; (2) we continue training until the validation loss flattens or a given computational budget is reached; and (3) for benchmarks on smaller datasets, we report the results by averaging over five runs to ensure fairness.

**Experimental Setup.** All experiments in Sections 4.1 and 4.2 are implemented in Keras with JAX backend using the examples available at `keras.io/examples` with minor dataset-specific adjustments, e.g., modifying the number of classes, layers, etc. The generative language modelling experiment in Section 4.2 is an adaptation of Andrea Karpathy's NanoGPT available at `github.com/karpathy/nanoGPT`. All the reported results are obtained by training on an Nvidia RTX 4090 GPU (24GB VRAM) or an Nvidia A100 GPU (80GB VRAM); however, we have chosen model and batch sizes to ensure that they run on 24GB VRAM. In each table, we report the train and test loss and accuracy (where relevant), the number of parameters in one attention layer (in the "# Param." column), the average training time (in seconds) of models for one epoch on an RTX 4090 GPU (in the "Epoch Time" column), as well as other related task-specific metrics.

### 4.1 Vision Transformers

We experiment with three widely adopted vision datasets of varying size and complexity: MNIST (LeCun et al., 2010), CIFAR100 (Krizhevsky, 2009), and ImageNet1K Russakovsky et al. (2015). For Brevity, we refer to the ImageNet1K dataset throughout the paper as ImageNet. However, for the reported ImageNet results, we first pre-trained the model on the ImageNet21K dataset. We report the training details in Appendix A.3.

Table 1: Averages of different metrics (over five runs on MNIST and CIFAR100, and one run on ImageNet). The numbers in parentheses indicate the ranking of each mechanism on each dataset for that metric. An ablation study on the number of heads is available in Appendix A.3. An additional ablation study for models of the same size on ImageNet but with different attention mechanisms is provided in Appendix A.3. As expected, Efficient Attention models have the smallest attention layer size, and the Super Attention models achieve the highest accuracy and lowest loss.

| Dataset | Att. | $h$ | $d_m$ | # Param. | Epoch Time | Acc. (%) | Loss | Top 5 | Val Acc. (%) | Val Loss | Val Top 5 |
|---|---|---|---|---|---|---|---|---|---|---|---|
| MNIST | Stn. | 4 | 128 | 66K (4) | 8.31 (4) | 93.73 (4) | 0.209 (4) | N/A | 98.12 (4) | 0.062 (4) | N/A |
| | Opt. | 4 | 128 | 49K (3) | 7.68 (3) | 95.36 (2) | 0.161 (2) | N/A | 98.43 (2) | 0.046 (2) | N/A |
| | Eff. | 4 | 128 | **33K (1)** | **7.05 (1)** | 94.28 (3) | 0.197 (3) | N/A | 98.27 (3) | 0.058 (3) | N/A |
| | Sup. | 4 | 128 | 37K (2) | 7.58 (2) | **96.96 (1)** | **0.112 (1)** | N/A | **98.62 (1)** | **0.051 (1)** | N/A |
| CIFAR100 | Stn. | 8 | 256 | 263K (4) | 21.19 (4) | 72.28 (2) | 1.41 (2) | 91.02 (2) | 48.14 (3) | 1.82 (2) | 90.22 (4) |
| | Opt. | 8 | 256 | 197K (3) | 20.39 (3) | 72.26 (3) | 1.47 (3) | 93.01 (3) | 48.63 (2) | 1.71 (2) | 90.99 (2) |
| | Eff. | 8 | 256 | **131K (1)** | **19.22 (1)** | 71.96 (4) | 1.49 (4) | 92.23 (4) | 47.95 (4) | 1.83 (4) | 90.48 (3) |
| | Sup. | 8 | 256 | 197K (3) | 20.28 (2) | **79.62(1)** | **1.28 (1)** | **94.34 (1)** | **49.28 (1)** | **1.55 (1)** | **91.69 (1)** |
| ImageNet | Stn. | 12 | 768 | 2.36M (4) | 2572 (4) | 92.07 (2) | 1.02 (2) | 98.41 (2) | 74.35 (3) | 1.47 (3) | 94.10 (4) |
| | Opt. | 12 | 768 | 1.77M (3) | 2426 (2) | 91.78 (3) | 1.03 (3) | 98.36 (3) | 77.12 (2) | 1.47 (3) | 94.21 (3) |
| | Eff. | 12 | 768 | **1.18M (1)** | **2374 (1)** | 90.36 (4) | 1.05 (4) | 98.37 (4) | 75.67 (4) | 1.44 (2) | 95.46 (2) |
| | Sup. | 12 | 768 | 1.22M (2) | 2483 (3) | **94.09 (1)** | **0.94 (1)** | **99.32 (1)** | **79.29 (1)** | **1.39 (1)** | **96.37 (1)** |

**ViT Results Analysis.** The number of parameters in the models considered for the vision tasks range from 300K (MNIST) to 60M (ImageNet), their context length ranges from 64 (MNIST) to 256 (CIFAR100 and ImageNet), the dataset sizes range from 60K (MNIST) to 1.28M (ImageNet), and the number of classes ranges from 10 (MNIST) to 1K (ImageNet). We observe that in these all experiments, Super Attention performs better than all other attention mechanisms despite having fewer parameters than standard attention. Also, Optimized and Efficient Attention demonstrate comparable performance despite having fewer parameters than standard attention.

4.2 NATURAL LANGUAGE PROCESSING

In this section, we evaluate the attention variants considered here in Transformer models of different sizes for three NLP tasks: sentiment classification, Natural Machine Translation (NMT), and generative language modelling. For sentiment classification (Table 2), we use two widely-used benchmarks, IMDB Movie Reviews (Maas et al., 2011) and Amazon Reviews (Ni et al., 2019) datasets. For NMT (Table 3), we use the combined Europarl (Koehn, 2005) and Anki (Anki.net) dataset for English-to-Spanish translation. For generative language modelling (Table 4), we use the OpenWeb-Text dataset (Gokaslan & Cohen, 2019) for training and the HellaSwag dataset (Zellers et al., 2019) for comparing the common-sense reasoning performance of the trained models.

Table 2: Averages of different metrics over five runs in the natural language classification experiments on IMDB and Amazon Reviews datasets. The numbers in parentheses indicate the ranking of each attention variant for that metric for each dataset. Ablation studies on the number of heads for all experiments is available in Appendix A.4. Efficient Attention models have the smallest attention layer size and the Super Attention models perform the best in terms of accuracy and loss.

| Dataset | Att. | $h$ | $d_m$ | # Param. | Epoch Time | Acc. (%) | Loss | Val Acc. (%) | Val Loss |
|---|---|---|---|---|---|---|---|---|---|
| IMDB | Stn. | 4 | 32 | 4,224 (4) | 0.315 (4) | 95.70 (4) | 0.086 (3) | 77.62 (4) | 0.474 (4) |
| | Opt. | 4 | 32 | 3,168 (2) | 0.305 (3) | 96.31 (3) | 0.095 (4) | 77.85 (2) | 0.472 (2) |
| | Eff. | 4 | 32 | **2,112 (1)** | **0.280 (1)** | 96.41 (2) | **0.064 (1)** | 77.77 (3) | **0.468 (1)** |
| | Sup. | 4 | 32 | 3,168 (2) | 0.299 (2) | **97.45 (1)** | 0.070 (2) | **78.34 (1)** | 0.472 (2) |
| Amazon | Stn. | 4 | 64 | 16,640 (4) | 20.38 (4) | 62.54 (3) | 0.868 (3) | 52.74 (4) | 1.097 (4) |
| | Opt. | 4 | 64 | 12,480 (2) | 19.89 (3) | 61.64 (4) | 0.876 (4) | 52.88 (3) | 1.090 (3) |
| | Eff. | 4 | 64 | **8,320 (1)** | **17.20 (1)** | 63.55 (2) | 0.845 (2) | 53.19 (2) | 1.080 (2) |
| | Sup. | 4 | 64 | 12,480 (2) | 19.77 (2) | **66.52 (1)** | **0.774 (1)** | **54.25 (1)** | **1.058 (1)** |

**NLP Results Analysis.** The number of parameters in the models considered for the NLP tasks ranges from 650K (IMDB) to 124M (language modelling), their context length ranges from 32 (IMDB) to 1024 (language modelling), the dataset sizes range from 50K samples (IMDB) to 9 billion tokens (OpenWebText). We observe a similar pattern to ViT for text classification in the sense that

Table 3: Averages of different metrics over five runs for English-to-Spanish NMT on combined Europarl and Anki translation datasets. The numbers in parentheses indicate the ranking of each attention variant for that metric. Ablation studies on the number of heads for all experiments is available in Appendix A.4. Optimized and Efficient Attentions perform similarly to standard attention on most metrics with ½ and ¾ as many attention parameters, respectively. As the Super Attention layer has a fixed context length and the decoder requires a varying context length, using Super Attention would require us to use a sliding window, which would not be comparable to the full attention used for the other attention variants.

| Att. | $h$ | $d_m$ | $d_k$ | # Param. | Epoch Time | BLEU | Acc. | Loss | Val BLEU | Val Acc. | Val Loss |
|------|-----|-------|-------|----------|-----------|------|------|------|----------|----------|----------|
| Stn. | 4 | 1024 | 256 | 4.2M (3) | 600.0 (3) | 23.1 (2) | 81.11 (3) | 0.83 (3) | **22.8 (1)** | 81.41 (3) | 0.84 (3) |
| Opt. | 4 | 1024 | 256 | 3.1M (2) | 586.8 (2) | **24.5 (1)** | **82.06 (1)** | **0.78 (1)** | 22.6 (3) | **81.98 (1)** | **0.80 (1)** |
| Eff. | 4 | 1024 | 256 | **2.1M (1)** | **523.0 (1)** | 22.6 (3) | 81.15 (2) | 0.82 (2) | 22.3 (3) | 81.44 (2) | 0.83 (2) |

Super Attention outperforms attention variants in terms of train accuracy (up to $^{(66.52-62.64)}/_{62.64} = 6.19\%$ compared to standard attention on Amazon Reviews). We also observe that standard attention is slower than all other variants (up to $^{(600-523)}/_{523} = 14.72\%$ slower than Efficient Attention in NMT) with the highest number of parameters (twice as many parameters per layer compared to Efficient Attention). The generative language modelling experiment reveals subtle differences in performance among the models. The standard attention-based model demonstrated marginally lower training and validation losses compared to the Optimized Attention model. In turn, the Optimized Attention model slightly outperformed the Efficient Attention model in terms of loss. However, when evaluated on the HellaSwag benchmark, all three models exhibited comparable performance, achieving accuracy rates between 30% and 31%.

Table 4: Averages of different metrics in generative language modelling using NanoGPT, a widely-referenced re-implementation of GPT-2 124M by Andrea Karpathy, when using different attention architectures. The models are trained on the OpenWebText dataset (∼9B training tokens) for one epoch with a batch size of 500 and a micro-batch size of 5 using a single A100 80GB node. The maximum sequence length is set to 1024. In addition to the loss metric, we have provided the size of each model as well as the results of the evaluation on the HellaSwag benchmark. Similarly to the NMT task, a fair comparison of Super Attention against the other variants is not feasible as NanoGPT uses full attention but Super Attention requires using a sliding window.

| Att. | $h$ | $d_m$ | $d_k$ | Layer Size | Model Size | Train Loss | Val Loss | HellaSwag |
|------|-----|-------|-------|-----------|-----------|-----------|----------|-----------|
| Stn. | 12 | 768 | 64 | 2.36M | 124M | 2.92 | 3.13 | 0.31 |
| Opt. | 12 | 768 | 64 | 1.77M | 117M | 2.96 | 3.14 | 0.31 |
| Eff. | 12 | 768 | 64 | 1.18M | 110M | 3.02 | 3.18 | 0.30 |

## 4.3 SPEED AND FLOPS ANALYSIS

Appendices A.1 and A.2 are dedicated to studying the computational complexity and inference speed of the considered attention variants. Equation (24) formulates the computational complexity for each algorithm. Figure 2 visualizes a comparison between the required number of FLOPs for each algorithm based on "sequence length" and "projection dimension". It indicates Efficient Attention requires the least number of FLOPs under all scenarios. From an empirical perspective, Table 5 and Figure 3 exhibit the faster inference speed (lower latency) of Efficient Attention compared to other variants in different datasets, followed by Optimized and Super Attention variants.

## 5 RELATED WORK

After the adoption of Transformers, different research directions have emerged to address different shortcomings of the attention mechanism and Transformer models. Sparse attention, such as Longformer (Beltagy et al., 2020; Zhang et al., 2021a), reduces the computational complexity by focusing on key input parts (Child et al., 2019). Despite their efficiency in handling long sequences, sparse attention mechanisms struggle with tasks requiring a comprehensive sequence analysis.

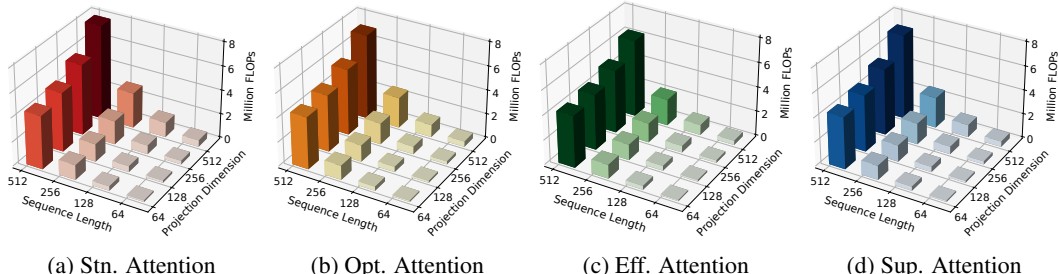

(a) Stn. Attention     (b) Opt. Attention     (c) Eff. Attention     (d) Sup. Attention

Figure 2: 3D plots visualizing the number of required FLOPs for each attention variant during a forward plus backward pass given different sequence lengths and projection dimensions in a single head setting. Efficient Attention followed by Super Attention and Optimized Attention needs significantly fewer FLOPs for completing a forward and backward pass compared to standard attention.

Another line of research focuses on approximating the attention matrix to attain linear complexity. Performer (Choromanski et al., 2021) uses random feature maps and FAVOR+ mechanism; Linformer (Wang et al., 2020) projects keys and values to lower dimensions by exploiting low-rank properties of attention matrices. While effective for long sequences, using approximation strategies often leads to reduced model quality compared to calculating exact attention, particularly for tasks requiring precise token relationships.

A new line of research focuses on architectures that combine transformers' parallel training speed with RNNs' inference efficiency. These include RWKV (Peng et al., 2023), which uses linear recurrence and learnable time-mixing parameters, and State-Space models like S4 (Gu et al., 2021) and Mamba (Gu & Dao, 2024), which leverage structured state-space sequences for long-range dependencies. While these approaches show promise through efficient inference and strong theoretical properties, Transformers maintain dominance due to their proven scalability in large language models and superior performance on parallel hardware during training.

Transformers' dominance has prompted a line of research for addressing their inefficiencies. For instance, Voita et al. (2019) show that multi-head SDPA is over-parameterized and the majority of heads can be pruned without negatively affecting the performance. Using this insight, Cordonnier et al. (2020) introduce a collaborative framework for reducing the size of key and query projections significantly without performance degradation.

Sparsification techniques reduce the number of non-zero elements in a network's weights. Ashkboos et al. (2024) introduced a post-training sparsification technique for large language models that compresses weight matrices with 1-10% performance degradation. Increasing sparsity could lead to reduced robustness (Timpl et al., 2022). In addition to these directions, we discuss further related attempts (including research on LoRA, Quantization and Flash Attention) for facilitating the deployability of transformer models in Appendix B.

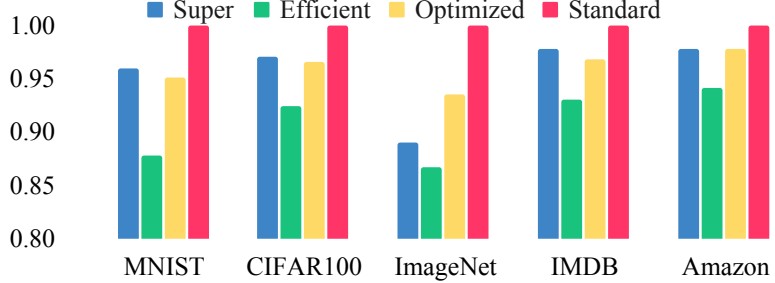

Figure 3: Summary of relative inference latency of the models using different attention variants relative to standard attention on different datasets on an Edge Device (Apple Laptop M2 Chip). Efficient Attention is the fastest while Optimized and Super Attention are also faster than standard attention. More details and numerical results for all datasets are available in Table 5.

## 6 DISCUSSION

We proposed and evaluated three variants of SDPA, which alter the standard arrangement of linear transformations in standard SDPA in order to uncover if by doing so, we can achieve better performance per computation cost and number of parameters. These variants include Optimized, Efficient, and Super Attention (see Figure 1 for visualizations of each of the variants). Efficient and Optimized Attentions considerably reduce the size and computational cost of the attention layer, while performing reasonably close to standard attention. Super Attention performs better than all other variants despite having fewer parameters than standard attention. More precisely, our experimental results can be summarized as follows.

**Computer Vision.** We considered image classification on MNIST, CIFAR100, and ImageNet1K, comparing standard, Optimized, Efficient, and Super Attention. In terms of performance, i.e., accuracy and loss, Optimized and Efficient Attention performed similarly to standard attention, while having fewer parameters, and being faster to train and infer. Super Attention outperforms standard attention in terms of accuracy by 3.5%, 10.1%, and 2.2% on MNIST, CIFAR100, and ImageNet datasets, respectively, while being smaller and faster to train and infer.

**Natural Language Processing.** We also considered a wide range of NLP tasks, including sentiment classification on IMDB Movie Reviews and Amazon Reviews, NMT on combined Europarl Parallel Corpus and Anki datasets for English-to-Spanish translation, and generative language modelling on OpenWebText dataset. Optimized and Efficient Attention performed similarly to standard attention on all tasks while having fewer parameters and being faster, and Super Attention outperforms standard attention by 1.8% and 6.4% on IMDB and Amazon Reviews respectively.

**Limitations.** There are two limitations in this paper. First, Super Attention supports fixed context length due to the fixed size of $W^A$ (see Equation (22) and Figure 1d). Nonetheless, these do not affect the advantages of Super Attention in many SotA applications such as in ViT. Moreover, this can be addressed using a sliding window, which a future work is currently in progress. Second, because of limited computational resources, we could only validate our hypotheses on models with up to 124 million (1.1 billion considering Appendix A.5) parameters trained on datasets with up to 9 billion (30 billion considering Appendix A.5) tokens. Further scaling the experiments beyond our computational resources and training large multi-modal and language models using the proposed mechanisms could facilitate a better understanding of their performance in industrial scales.

## 7 CONCLUSIONS

We investigated SDPA and three proposed variants that modify the arrangement of linear transformations in SDPA. Two variants, Optimized and Efficient Attention, replace one (values) and two (values and keys) linear transformations in SDPA with slicing, resulting in 25% and 50% size reductions and fewer matrix multiplications, respectively. The third variant, Super Attention, introduces a new linear transformation operating on the values from the left. While Super Attention can be applied to standard, Optimized, or Efficient Attention, we focused on combining it with Efficient Attention to reduce the number of parameters in the attention layer, resulting in approximately 25% fewer parameters compared to standard attention.

We evaluated all discussed variants across a wide range of tasks (within our available computational budget), from image classification to generative language modelling, using benchmarks varying in size from 60,000 examples to 9 billion tokens. Our evaluations demonstrate Optimized Attention and Efficient Attention perform comparably to standard attention across different benchmarks, despite having considerably fewer parameters. Super Attention outperforms all variants in all applicable benchmarks while still maintaining fewer parameters than standard attention. In summary, the proposed attention variants show promising performance across a wide range of tasks. Our generative language modelling experiment using a 1.1B Llama-based model in Appendix A.5 provides some insight into their performance on large scales. Realizing the true potential of these variants requires evaluating larger scales, which are beyond our available resources. Overall, the promising results of the proposed variants suggest the potential for more extensive evaluation and adoption.

**Reproducibility Statement.** The code for all experiments is provided in the supplementary materials. Publicly available datasets are used, with automatic downloads included in the code, except for the Amazon dataset (link in README). The NanoGPT repository (linked in Experimental Setup) details the generative language modelling experiment. Further implementation details are in Section Section 4 and Appendices A.3 and A.4.

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

# A  ADDITIONAL EXPERIMENTS

## A.1  EDGE DEVICE PERFORMANCE

Our main motivation for introducing Optimized, Efficient, and Super Attention is to allow running more capable models on edge devices. We calculated the inference times of the Transformer models, we trained before, on a MacBook Pro with an M2 Chip for each task/attention mechanism in Table 5. As expected, Efficient models are the fastest. Also, Super Attention and Optimized Attention models are faster than their standard counterparts with the same number of heads while performing equally well as we discussed before.

Table 5: Total inference times (in seconds) for each attention mechanism/dataset pair on an Apple M2 chip over 5,000 samples.

| Name | $h$ | MNIST | CIFAR100 | ImageNet | IMDB | Amazon |
|---|---|---|---|---|---|---|
| Standard | 1 | 4.43 | 34.84 | 299.26 | 0.114 | 1.02 |
| | 4 | 5.27 | 46.06 | 323.84 | 0.183 | 1.77 |
| | 8 | 6.89 (4) | 62.08 (4) | 341.69 (4) | 0.266 (4) | 2.84 (4) |
| Optimized | 1 | 4.19 | 33.36 | 281.14 | 0.109 | 1.00 |
| | 4 | 5.22 | 44.17 | 301.30 | 0.176 | 1.72 |
| | 8 | 6.37 (2) | 60.63 (2) | 320.49 (3) | 0.262 (2) | 2.77 (2) |
| Efficient | 1 | 3.78 | 31.50 | 259.71 | 0.101 | 0.93 |
| | 4 | 4.71 | 42.16 | 276.15 | 0.170 | 1.66 |
| | 8 | **6.10 (1)** | **58.60 (1)** | **301.24 (1)** | **0.256 (1)** | **2.70 (1)** |
| Super | 1 | 4.21 | 33.69 | 264.99 | 0.112 | 0.99 |
| | 4 | 5.07 | 44.47 | 284.49 | 0.178 | 1.74 |
| | 8 | 6.65 (3) | 60.73 (3) | 309.72 (2) | 0.264 (3) | 2.77 (2) |

## A.2  SPEED AND EFFICIENCY COMPARISON

In the main body and other sections of the Appendix, we present comprehensive theoretical comparisons and rigorous experiments on Vision and NLP classification tasks as well as for English-to-Spanish translation to compare the attention algorithms. Optimized Attention and Efficient Attention perform on par with standard attention with 25% and 50% less parameters respectively. In addition, Super Attention outperformed all other algorithms significantly while having 25% fewer parameters compared to standard attention.

As mentioned in the main body, according to the definitions of our proposed algorithms, Efficient, Optimized, and Super Attention mechanisms perform 2,1, and 1 fewer matrix multiplication per head compared to standard attention respectively. Here, we further analyze and compare the required number of FLOPs for completing a single forward and backward pass for all algorithms under study to gain further insight into the efficiency of the proposed algorithms.

**FLOPs Versus Projection Dim.**  As depicted in Figure 4, we compare the number of required FLOPs by each attention algorithm when we fixate the sequence length (denoted as $\ell$) and vary the projection dimension. Even though the number of FLOPs scales linearly with the projection dimension for all algorithms, the slope of this increase differs significantly for each algorithm. Specifically, for Efficient Attention, the slope of the line is equal to $9\ell$ while for both Optimized and Super Attention this is equal to $12\ell$ compared to $15\ell$ for standard attention. This means that as we scale the projection dimension the FLOPs required for finishing a forward and backward pass using Efficient Attention increases ⅗ as fast as standard attention.

**FLOPs Equation.**  The number of FLOPs required for finishing a forward and backward pass for each of the attention mechanisms is calculated according to the following equation:

$$\text{FLOPs} = C_{\text{Attn}}\ell d_m + 15h\ell^2 \tag{24}$$

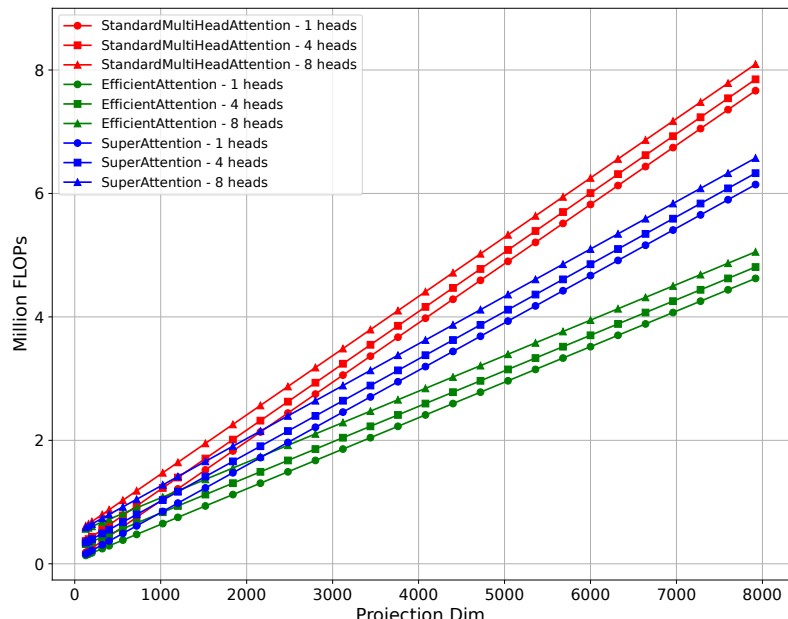

Figure 4: Number of Flops required to complete a single forward plus backward pass for each attention mechanism. While the complexity and therefore, the number of FLOPs increases linearly as the projection dimension increases for all attention mechanisms, the slope of the increase varies significantly as depicted in this plot. Efficient Attention and Super Attention (Optimized Attention is not shown as it is exactly similar to Super Attention) require significantly fewer FLOPs as the projection dimension increases compared to standard attention. Here sequence length is set to 64 ($\ell = 64$). Trying different values for $\ell$ changes the scale of the $y$-axis but the chart looks the same.

where $C_{\text{Attn}}$ is the attention algorithm constant which is 15 for standard attention, 12 for Optimized and Super Attention, and 9 for Efficient Attention, and $\ell$, $d_m$, and $h$ represent the sequence length, projection dimension, and number of heads consistent with the notation used throughout the paper.

Figure 2 shows the 3D plot summarizing the number of FLOPs for each attention algorithm under varying sequence length and projection dimension in the single head setting. As evident in Figure 2 and Equation (24), our proposed algorithms need fewer FLOPs as sequence length increases, which is an important consideration for use in LLMs.

**FLOPs Heatmaps.** In addition to the previous analyses, in Figure 5, we compare the ratio of FLOPs required to finish a single forward and backward pass by standard attention to Efficient Attention under different settings (i.e., varying sequence length and projection dimension) for different number of heads. In all scenarios, standard attention requires up to 66% more FLOPs in comparison to Efficient Attention. On average, Standard Efficient requires 30%, 25%, 20%, and 16% more FLOPs in comparison to Efficient Attention when using 1, 2, 4, and 8 heads, respectively.

### A.3 VISION TRANSFORMERS

**MNIST.** We trained ViT models with different attention mechanisms, all with two attention layers and model dimension $d_m = 128$. As expected, Super Attention outperforms all other architectures, in terms of accuracy, by at least $2.68\%$ and standard attention by $3.23\%$. The smallest attention layer size belongs to Efficient Attention, which performs on par with standard attention. The complete results are presented in Table 6.

**ImageNet.** Scaling the vision experiments even further, the ImageNet1k dataset presents much more complexity as the labels comprise 1000 classes. We used a modified ViT-B/16 model architecture, employed different attention mechanisms in its Transformers blocks, and trained the models. Due to our computational constraints, we reduced the number of transformer blocks from 12 to 8,

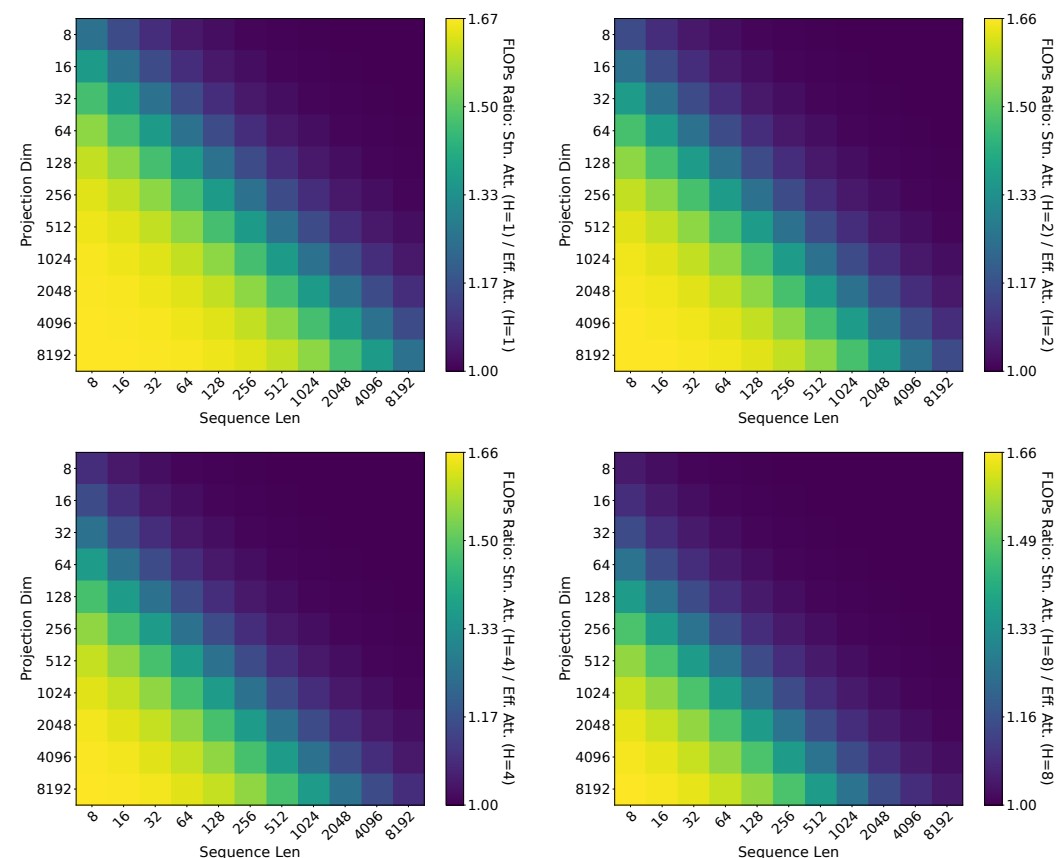

Figure 5: Heatmaps showing the ratio of FLOPs Standard Attention requires compared to the Efficient Attention in 1, 2, 4, and 8 attention head settings. Standard attention requires up to 67% more FLOPs to complete a single forward and backward pass. On average, standard attention requires 30%, 25%, 20%, and 16% more FLOPs than Efficient Attention when using 8, 4, 2, and 1 heads respectively.

resized the images to $112 \times 112$ (instead of the original $224 \times 224$) and reduced the patch size from 16 to 8 to enable training on our Nvidia RTX 4090 GPU. Other parameters are similar to the original architecture; specifically, $d_m = 768$ and $h = 12$. Tables 1 and 7 present the results of our experiments on the ImageNet dataset.

Val. results in Tables 1, 6 and 7 refer to models' performances on the official validation set for ImageNet1K, and the official tests sets for MNIST and CIFAR100 datasets.

## A.4 NATURAL LANGUAGE PROCESSING

### A.4.1 TRANSFORMER FOR TEXT CLASSIFICATION

**IMDB.** The IMDB dataset includes 50,000 reviews with binary labels, indicating negative and positive sentiments. The Transformer models, used in this experiment, all have a single attention layer with model dimension and context length 32. The complete results are presented in Table 8.

**Amazon Reviews.** The Amazon Reviews dataset poses a different challenge than the IMDB dataset as it is a significantly larger dataset with 3,650,000 reviews, containing a wider range of sentiments in $1, 2, \ldots, 5$; higher values indicate more positive sentiment. The Transformer models, used in this experiment, all have three attention layers with model dimension and context length 64. The complete results are presented in Table 9.

Table 6: Averages of different metrics over five runs in the MNIST experiment. The numbers in parentheses indicate the ranking of each mechanism for that metric. An ablation study on the number of heads shows increasing the number of heads enhances the performance of all algorithms. As expected, the Efficient Attention model has the smallest attention layer size and the Super Attention model performs the best in terms of accuracy and loss.

| Att. | $h$ | $d_m$ | $d_k$ | # Param. | Avg. Time (s) | Acc. (%) | Loss | Val Acc. (%) | Val Loss |
|------|-----|-------|-------|----------|---------------|----------|------|--------------|----------|
|      | 1 | 128 | 128 | 66,048 | 8.15 | 93.26 | 0.227 | 98.02 | 0.063 |
| Stn. | 2 | 128 | 64 | 66,048 | 8.18 | 95.40 | 0.161 | 98.61 | 0.049 |
|      | 4 | 128 | 32 | 66,048 (4) | 8.31 (4) | 93.73 (4) | 0.209 (4) | 98.12 (4) | 0.062 (4) |
|      | 1 | 128 | 128 | 49,536 | 7.56 | 91.02 | 0.299 | 97.30 | 0.095 |
| Opt. | 2 | 128 | 64 | 49,536 | 7.57 | 93.70 | 0.215 | 97.93 | 0.071 |
|      | 4 | 128 | 32 | 49,536 (3) | 7.68 (3) | 95.36 (2) | 0.161 (2) | 98.43 (2) | 0.046 (2) |
|      | 1 | 128 | 128 | 33,024 | 6.89 | 93.29 | 0.228 | 97.78 | 0.073 |
| Eff. | 2 | 128 | 64 | 33,024 | 6.99 | 93.60 | 0.223 | 98.11 | 0.061 |
|      | 4 | 128 | 32 | **33,024 (1)** | **7.05 (1)** | 94.28 (3) | 0.197 (3) | 98.27 (3) | 0.058 (3) |
|      | 1 | 128 | 128 | 37,184 | 7.46 | 96.24 | 0.136 | 98.32 | 0.056 |
| Sup. | 2 | 128 | 64 | 37,184 | 7.50 | 96.59 | 0.124 | 98.52 | 0.050 |
|      | 4 | 128 | 32 | 37,184 (2) | 7.58 (2) | **96.96 (1)** | **0.112 (1)** | **98.62 (1)** | **0.051 (1)** |

Table 7: Performance of different architectures on the ImageNet dataset. Since different attention layer architectures in the main ImageNet experiment had different numbers of parameters, an interesting ablation study is comparing these architectures when the total number of parameters is very close. To achieve this, we change some hyperparameters like $d_m$ or the number of attention layers from the previous experiment. The numbers in parentheses indicate the ranking of each mechanism for that metric. We used a modified ViT-B/16 model, plugged in the attention algorithms in the Transformers block, and trained the models. Super Attention significantly outperforms all other algorithms. Unlike the results reported in Table 1 in the main body, the models in this ablation experiment are not pre-trained on ImageNet21K (as such the accuracies and validation accuracies are lower compared to the ones with pre-training).

| Att. | $h$ | $d_m$ | Att. Layers | Tot. # Param. | Acc. (%) | Loss | Top 5 | Val Acc. (%) | Val Loss | Val Top 5 |
|------|-----|-------|-------------|---------------|----------|------|-------|--------------|----------|-----------|
| Stn. | 12 | 768 | 8 | 60.54M (4) | 51.18 (4) | 2.09 (4) | 76.05 (4) | 32.74 (4) | 3.36 (4) | 56.48 (4) |
| Opt. | 12 | 816 | 8 | 60.12M (2) | 53.22 (2) | 1.98 (2) | 77.21 (2) | 33.44 (3) | 3.23 (3) | 57.37 (3) |
| Eff. | 12 | 804 | 9 | **60.09M (1)** | 51.28 (3) | 2.06 (3) | 76.66 (3) | **35.49 (1)** | **3.13 (1)** | **59.69 (1)** |
| Sup. | 12 | 804 | 9 | 60.44M (3) | **64.98 (1)** | **1.37 (1)** | **87.36 (1)** | 34.31 (2) | 3.18 (2) | 58.70 (2) |

### A.4.2 TRANSFORMER FOR NEURAL MACHINE TRANSLATION

**Europarl Parallel Corpus and Anki.** Anki dataset for English-Spanish translation consists of more than 118,000 sentence pairs in both English and Spanish languages. While training a model on this dataset enables basic translation, the educational nature and size of the dataset are too simple for training a capable translation model. Therefore, we also add the Europarl Parallel Corpus which has around 2 million examples in both English and Spanish languages and has sentences with much more technical and sophisticated terms to enable training in a powerful English-to-Spanish translation model. We then shuffle the mix of both datasets, and randomly split the dataset into 99.8%, 0.1%, and 0.1% for train, validation, and test splits respectively.

We then train a translation model inspired by the implementation available on the official Keras website for translation but with 2 decoder blocks and one encoder block for 6 epochs. Additionally, we set the $d_m = 1024$ and try 1, 2, and 4 as the number of heads. We use Sparse Categorical Cross Entropy as our loss metric. The complete analysis of the results is available in Table 10.

All 3 algorithms perform comparably in terms of BLEU score, Accuracy, and Loss. However, the number of attention parameters per encoder/decoder layer is ½ and ¾ of standard attention in Efficient and Optimized Attention respectively. Additionally, Efficient attention is up to $^{(556.5-472.7)}/_{556.6} = 15.06\%$ faster to train in comparison to the standard attention.

Table 8: Averages of different metrics over five runs in the IMDB experiment. Here, varying the number of heads doesn't meaningfully affect the performance of any of the algorithms. As expected, the Efficient Attention model has the smallest attention layer size and the Super Attention model performs the best in terms of accuracy and loss.

| Att. | $h$ | $d_m$ | $d_k$ | # Param. | Avg. Time | Acc. (%) | Loss | Test Acc. (%) | Test Loss |
|---|---|---|---|---|---|---|---|---|---|
| | 1 | 32 | 32 | 4,224 | 0.284 | 96.09 | 0.082 | 78.09 | 0.461 |
| Stn. | 2 | 32 | 16 | 4,224 | 0.297 | 95.51 | 0.112 | 78.14 | 0.467 |
| | 4 | 32 | 8 | 4,224 (4) | 0.315 (4) | 95.70 (4) | 0.086 (3) | 77.62 (4) | 0.474 (4) |
| | 1 | 32 | 32 | 3,168 | 0.283 | 96.62 | 0.070 | 78.00 | 0.461 |
| Opt. | 2 | 32 | 16 | 3,168 | 0.299 | 96.77 | 0.073 | 78.00 | 0.460 |
| | 4 | 32 | 8 | 3,168 (2) | 0.305 (3) | 96.31 (3) | 0.095 (4) | 77.85 (2) | 0.472 (2) |
| | 1 | 32 | 32 | 2,112 | 0.267 | 96.66 | 0.080 | 77.58 | 0.478 |
| Eff. | 2 | 32 | 16 | 2,112 | 0.273 | 96.86 | 0.068 | 77.74 | 0.473 |
| | 4 | 32 | 8 | **2,112 (1)** | **0.280 (1)** | 96.41 (2) | **0.064 (1)** | 77.77 (3) | **0.468 (1)** |
| | 1 | 32 | 32 | 3,168 | 0.272 | 97.68 | 0.063 | 78.21 | 0.472 |
| Sup. | 2 | 32 | 16 | 3,168 | 0.294 | 97.84 | 0.064 | 78.35 | 0.454 |
| | 4 | 32 | 8 | 3,168 (2) | 0.299 (2) | **97.45 (1)** | 0.070 (2) | **78.34 (1)** | 0.472 (2) |

Table 9: Averages of different metrics over five runs in the Amazon Reviews experiment. An ablation study on the number of heads shows increasing the number of heads helps improve the performance of all algorithms. The Efficient Attention model has the smallest attention layer size and the Super Attention model performs the best in accuracy and loss.

| Att. | $h$ | $d_m$ | $d_k$ | # Param. | Avg. Time | Acc. | Loss | Val Acc. | Val Loss |
|---|---|---|---|---|---|---|---|---|---|
| | 1 | 64 | 64 | 16,640 | 13.81 | 61.33 | 0.897 | 52.84 | 1.094 |
| Stn. | 2 | 64 | 32 | 16,640 | 16.33 | 63.61 | 0.851 | 52.71 | 1.091 |
| | 4 | 64 | 16 | 16,640 (4) | 20.38 (4) | 62.54 (3) | 0.868 (3) | 52.74 (4) | 1.097 (4) |
| | 1 | 64 | 64 | 12,480 | 12.54 | 60.71 | 0.909 | 52.79 | 1.093 |
| Opt. | 2 | 64 | 32 | 12,480 | 14.67 | 62.04 | 0.884 | 52.93 | 1.090 |
| | 4 | 64 | 16 | 12,480 (2) | 19.89 (3) | 61.64 (4) | 0.876 (4) | 52.88 (3) | 1.090 (3) |
| | 1 | 64 | 64 | 8,320 | 10.61 | 62.23 | 0.873 | 53.25 | 1.082 |
| Eff. | 2 | 64 | 32 | 8,320 | 14.05 | 63.11 | 0.862 | 52.67 | 1.098 |
| | 4 | 64 | 16 | **8,320 (1)** | **17.20 (1)** | 63.55 (2) | 0.845 (2) | 53.19 (2) | 1.080 (2) |
| | 1 | 64 | 64 | 12,480 | 11.96 | 66.65 | 0.776 | 53.87 | 1.070 |
| Sup. | 2 | 64 | 32 | 12,480 | 15.21 | 66.30 | 0.781 | 54.11 | 1.064 |
| | 4 | 64 | 16 | 12,480 (2) | 19.77 (2) | **66.52 (1)** | **0.774 (1)** | **54.25 (1)** | **1.058 (1)** |

## A.5 EVALUATION FOR USE IN LLMS

In addition to evaluating the standard SDPA and its variants for generative language modelling in a scale of around 125M parameters, we also trained a Language Model (LM) with 1.1B parameters based on Efficient Attention architecture to see the feasibility and scalability of this variant of SDPA in a large scale experiment. This Language Model achieves lower loss than the similarly-sized TinyLlama model, which is based on Standard Attention (details are provided in Table 11 below). We could not train more LMs based on other architectures due to our limited computational resources. The LM based on Efficient Attention was trained using a GPU credit donation that we used to train our LM over 8 weeks on 30 billion tokens of C4 dataset (Raffel et al., 2019) using a single A100 with 80GB of GPU.

Table 10: Averages of different metrics over five runs trained on Europarl and Anki English-to-Spanish translation datasets. The numbers in parentheses indicate the ranking of each mechanism for that metric. An ablation study on the number of heads shows increasing the number of heads enhances the performance of all algorithms. Optimized and Efficient Attentions perform on par or better than Standard Attention on most benchmarks with 1/2 and 3/4 as many attention parameters.

| Att. | $h$ | $d_m$ | $d_k$ | # Param. | Avg. Time | BLEU | Acc. | Loss | Val BLEU | Val Acc. | Val Loss |
|------|-----|-------|-------|----------|-----------|------|------|------|----------|----------|----------|
| | 1 | 1024 | 1024 | 4,198,400 | 556.5 | 23.2 | 80.48 | 0.86 | 22.1 | 80.86 | 0.87 |
| Stn. | 2 | 1024 | 512 | 4,198,400 | 598.7 | 22.3 | 81.03 | 0.84 | 22.7 | 81.43 | 0.84 |
| | 4 | 1024 | 256 | 4,198,400 (3) | 600.0 (3) | 23.1 (2) | 81.11 (3) | 0.83 (3) | **22.8 (1)** | 81.41 (3) | 0.84 (3) |
| | 1 | 1024 | 1024 | 3,148,800 | 552.0 | 22.5 | 81.15 | 0.87 | 22.6 | 81.11 | 0.84 |
| Opt. | 2 | 1024 | 512 | 3,148,800 | 583.8 | 22.1 | 81.61 | 0.82 | 23.0 | 81.57 | 0.82 |
| | 4 | 1024 | 256 | 3,148,800 (2) | 586.8 (2) | **24.5 (1)** | **82.06 (1)** | **0.78 (1)** | 22.6 (3) | **81.98 (1)** | **0.80 (1)** |
| | 1 | 1024 | 1024 | 2,099,200 | 472.7 | 22.4 | 81.13 | 0.82 | 22.8 | 81.43 | 0.83 |
| Eff. | 2 | 1024 | 512 | 2,099,200 | 498.6 | 22.3 | 81.48 | 0.80 | 22.9 | 81.62 | 0.81 |
| | 4 | 1024 | 256 | **2,099,200 (1)** | **523.0 (1)** | 22.6 (3) | 81.15 (2) | 0.82 (2) | 22.3 (3) | 81.44 (2) | 0.83 (2) |

Table 11: A Language Model (Based on Efficient Attention) compared to TinyLlama (Based on Standard Attention) after training on 30 billion tokens of C4 dataset. We set the number of heads to 1 in this LM to make training faster. Despite this, this LM performs favourably (5.8% smaller categorical cross-entropy loss) compared to TinyLlama.

| name | # layers | # heads | model dim | intermediate size | loss |
|------|----------|---------|-----------|-------------------|------|
| TinyLlama | 22 | 32 | 2048 | 5632 | 2.25 |
| Efficient based LM | 10 | 1 | 3072 | 8192 | 2.12 |

# B    ADDITIONAL RELATED WORK

Flash Attention (Dao et al., 2022) and Flash Attention 2 (Dao, 2024) optimize multi-head attention for modern GPUs without changing its structure, enabling faster processing and reduced memory demands. It's worth mentioning our proposed algorithms also benefit from these optimizations.

With the adoption of LLMs and Foundation Models (FMs), a lot of work has been done to improve their scalability and deployability. LoRA (Hu et al., 2022) adapts pre-trained models with minimal additional parameters, and QLoRA (Dettmers et al., 2023) incorporates quantization to reduce memory and computational demands.

Quantization has revolutionized the adoption of FMs, particularly those based on Transformers. Recent advances include mixed-precision post-training quantization for vision transformers (Liu et al., 2021), quantization-aware training (Jacob et al., 2018; Nagel et al., 2022), mixed-precision training (Micikevicius et al., 2018), dynamic quantization (Zhang et al., 2021b), and layer-wise quantization (Chen et al., 2019).

Moreover, Ding et al. (2022) unveiled a cutting-edge framework enhancing quantized model accuracy without significant performance degradation. However, quantization faces challenges such as potential performance drops and increased vulnerability to adversarial attacks (Hong et al., 2021; Gupta & Ajanthan, 2022).

