# OpenReview forum: "A Study of Necessity & Sufficiency of Linear Transformations in the Attention Mechanism"
_ICLR.cc/2025/Conference — Submitted to ICLR 2025_

### Official Review · Reviewer_VBYW · 2024-10-30

**Soundness:** 2
**Presentation:** 3
**Contribution:** 2
**Rating:** 3
**Confidence:** 5

**Summary:**

This paper introduces three efficient variants of Scaled Dot Product Attention (SDPA) that reduce computational costs and parameters by adjusting the linear transformations in them. Tested across vision and NLP tasks, these variants maintain or enhance performance while achieving faster inference.

**Strengths:**

1. The approach is simple and well-explained
2. Evaluation covers a broad range of tasks spanning different modalities.
3. Complexity analysis is comprehensive

**Weaknesses:**

I am not an NLP expert and am less familiar with NLP tasks, so my focus is primarily on the architecture and vision tasks.

1. Lack of Strong Baselines: The paper primarily compares against standard attention, which is a highly under-optimized baseline. For vision tasks, it would be more informative to include stronger, more optimized baselines such as Swin, MetaFormer, MaxViT, or EfficientNet. Since your method modifies the attention mechanism, it could be integrated into these attention-based or hybrid architectures, which would reveal how your proposed changes perform in comparison to already optimized attention modules.

2. Absence of End-to-End Training on ImageNet-1K: The goal of the proposed models seems to be achieving parameter efficiency without sacrificing expressiveness. However, fine-tuning on ImageNet-1K alone may not fully validate this. Superior performance on smaller datasets (e.g., CIFAR-100, MNIST) and fine-tuned ImageNet-1K can sometimes reflect lower expressiveness, as reduced architectures naturally avoid overfitting on simpler datasets. While removing components from the standard ViT could reduce redundancy, recent hybrid models like MetaFormer and NAT, which use standard ViT only in the later layers, demonstrate that the later layers often require more expressive power. Testing your method on such stronger baselines might show that similar gains are less achievable when the baseline model is already well-optimized and not overly redundant.

3. Limited Analysis Beyond Complexity: While the paper focuses on the complexity benefits of the proposed method, it would be insightful to explore how the learned representations change. For instance, comparing attention maps and MLP activations between standard attention and your optimized attention variants could reveal additional insights into the effects of these architectural modifications.

**Questions:**

N/A

---

> ### Author Response · Authors · 2024-11-21
>
> We thank reviewer VBYW for providing their feedback on some of the vision experiments in our paper.
>
> > 1. Lack of Strong Baselines ....
>
> The contributions of papers such as EfficientNet, MaxViT, MetaFormer, and Swin are **orthogonal** to our contribution. Specifically, **MetaFormer** uses a token mixer component instead of the attention module in transformers and shows that they are still doing quite well. **SWIN**'s contribution is not focused on the attention mechanism but rather on other concepts like dividing images into non-overlapping windows/patches and introducing periodic window shifting that enables cross-window connection using a hierarchical structure. **MaxViT** proposes a model-level architectural design that combines convolutional layers with global and local attention. **EfficientNet** focuses on parameter selection and design for convolutional neural networks and does **not** present any contribution for models using attention. Some of these contributions (SWIN and MaxViT) which are more model/block level contributions **can benefit from our work on attention mechanism**. However, they are not contributions that could reflect and study the merits of our contribution or be compared with our contributions which are layer-level architectures. Still, we believe future works building on top of those model-level architectures could use our contribution to build more efficient, stronger models.
>
> ---
>
> > 2. Absence of End-to-End Training on ImageNet-1K ....
>
> The **end-to-end training on ImageNet-1k is presented in the appendix of our paper**. Please see **Table 7 for end-to-end training results on ImageNet-1k**. As you probably know, when you train the models on ImageNet-1k only, it's impossible to reach validation and test accuracies above ~45% range, and this is why in our ImageNet experiment in the main body, we included results for models pre-trained on the 21k version. In a previous submission of this paper to NeurIPS, we had included only end-to-end training results and the reviewers asked us to see pre-trained models that can achieve higher validation and test accuracies as they wanted to know more about the generalizability of these models. This is why we have included these results and it’s really challenging/confusing for us as authors since reviewers have different preferences in this case and **to address the needs of both types, we have included both experiments, one in the main body and one in the appendix**. Please check the one in Table 7 that presents the experiment you are asking in your point.
>
> ---
>
> > 3. Limited Analysis Beyond Complexity: ...
>
> We have included a **comprehensive analysis of performance, time, and space complexity under a wide range of experiments, modalities and hyperparameter selection**. However, we understand you would like to see attention maps as well and to this end, we are now working on preparing attention maps as well for each architecture to strengthen our analysis. We will add this and submit the updated manuscript for the camera-ready version if the paper is accepted.

---

> ### Comment · Reviewer_VBYW · 2024-11-27
>
> Thank you for your clarification.
>
> 1. Lack of strong baselines
>
> I stand by my opinion on this point. To make it more clear, I believe this work introduces a method that can potentially be adapted to a wide range of models incorporating attention mechanisms, such as SWIN, MaxViT, and CaFormer (MetaFormer). However, whether the performance improvements observed with ViT will generalize to these advanced architectures remains uncertain. Based on the evidence provided, I am not convinced that this method offers a clear advantage over the standard attention mechanism in more sophisticated models. On the contrary, I suspect the observed improvements may stem from under-optimization in the ViT architecture rather than inherent superiority.
>
> 2. Accuracy on end-to-end ImageNet-1K
>
> I have some doubts regarding the reported accuracy range. From my experience, a 20M-parameter model can typically achieve around 80% accuracy on ImageNet-1K (refer to the lower 80% range models at: https://paperswithcode.com/sota/image-classification-on-imagenet). Could you clarify if I might be overlooking something specific in your experimental setup?

---

> ### Author Response · Authors · 2024-11-27
>
> Thank you for getting back to us with your feedback on our rebuttal.
>
>  > 1. Lack of strong baselines
> >
> > I stand by my opinion on this point. To make it more clear, I believe this work introduces a method that can potentially be adapted to a wide range of models incorporating attention mechanisms, such as Swin, MaxViT, and CaFormer (MetaFormer). ...
>
> We empathize with your concern. We agree with you that advancements such as Swin help improve the optimization and the performance of ViTs. Taking your concern into account, we started training Swin models using our proposed architectures on ImageNet-1k (end-to-end training) to address both your concerns about seeing results on more optimized ViT architectures and also another end-to-end training result on ImageNet-1k. We didn't expect we would have results ready by the previous discussion deadline. Now that the discussion period has been extended, we estimate the results will be ready by the weekend, and we will get back to you about the Swin experiment using our architectures.
>
>  > 2. Accuracy on end-to-end ImageNet-1K
> >
> > I have some doubts regarding the reported accuracy range. From my experience, a 20M-parameter model can typically achieve around 80% accuracy on ImageNet-1K (refer to the lower 80% range models at: https://paperswithcode.com/sota/image-classification-on-imagenet). Could you clarify if I might be overlooking something specific in your experimental setup?
>
> We understand your point. Let us consider the Swin models (as their Swin-T model has 29M parameters) and the
> reported results in the original Swin paper [1].
> For "regular end-to-end" training on the ImageNet1K dataset (see §4.1 in [1]), the paper mentions that they follow the training paradigm introduced in [2], which uses a teacher model that has been extensively trained on other datasets,
> and the target student model learns from the teacher model using a distillation token that ensures the student learns
> from the teacher through attention.
>
> This is different from pure end-to-end training on the ImageNet1K as the model is learning from a teacher
> that has been extensively trained. We cannot speak for all the ImageNet1K results reported on PapersWithCode, but all
> the papers we have investigated, which reach 70-80% accuracy, use either pretraining, distillation, or some other
> technique that uses knowledge learnt from other datasets. In the results reported in our paper, we have used
> pure end-to-end training for each dataset discussed (except for ImageNet, where we have reported the results with and without pre-training on ImageNet21K).
> In other words, we trained our models with randomly initialized
> weights directly on the dataset without any form of distillation or other techniques. However, for the new Swin
> experiment, which we will report soon, we follow the Swin paper [1], which uses the training paradigm introduced in [2].
>
> Meanwhile, as we are running the Swin experiment, please let us know if you have any questions or would like to discuss anything further.
>
> [1] *Swin Transformer: Hierarchical Vision Transformer using Shifted Windows*, Z. Liu, Yutong Lin, Y. Cao, H. Hu, Y. Wei, Z. Zhang, S. Lin, B. Guo, *ICCV 2021*
>
> [2] *Training Data-Efficient Image Transformers & Distillation through Attention*, H. Touvron, M. Cord, M. Douze, F. Massa, A. Sablayrolles, H. Jegou, *ICLR 2021*

---

> ### Author Response · Authors · 2024-12-01
>
> # Swin Benchmark (An End-to-End ImageNet Experiment)
>
> As you suggested, we improved our baselines for the ImageNet experiment by training different models using the Swin architecture following the training scheme used in the Swin paper. The results are summarised in the table below:
>
> **Table A. Evaluation of different architectures in Swin-S benchmark**
> |Att Type|Img. Size|#param.|Top-1 Acc. (%)| Param.-Acc. Trade-off. (%)|
> |---|---|---|---|---|
> |Stn. Att.| $224^2$ | 49.6M | 83.0 | 0.0 |
> |Opt. Att.|  $224^2$ |45.7M | 82.6| **+8.0** |
> |Eff. Att.|  $224^2$ | 41.7M | 82.1| **+17.6** |
> |Sup. Att.|  $224^2$ | 41.7M| 83.6 | **+19.8** |
>
> The last column in the table above measures
>
> $$\\frac{\\frac{Architecture\\ Accuracy}{Baseline\\ Accuracy}}{\\frac{Architecture\\ Parameter\\ Count}{Baseline\\ Parameter\\ Count}}.$$
>
> In other words, the numbers in the last column reflect that our models achieve better parameter-accuracy trade-offs compared to the baseline which is the Swin-S model using standard attention. The best-performing attention mechanism in this experiment is Super Attention, which achieves 0.6% higher accuracy compared to the Standard Swin-S model while having 16% fewer parameters.
>
> ---
>
> # Attention Maps
>
> As requested, we provide the attention maps for the end-to-end ImageNet1K experiments in the paper (Table 7). The attention maps can be viewed at the anonymous link below:
> https://anonymous.4open.science/r/ICLR_Rebuttal_Attention_Map-C52C
>
> We studied the attention map evolution across early, mid, and final transformer layers of ViTs in two examples. The overall pattern in the evolution of attention maps shows consistent behaviour across all models, with minor variations. The first layers in all models focus on regions encompassing entire object(s) in the image, demonstrated by attention to the complete dog in the first image and the babies in the cart in the second image. As we progress towards the mid and final layers, the attention maps become increasingly specialized, focusing on discriminative features such as babies' facial features or the dog's distinctive characteristics.
>
> While this evolutionary pattern is consistent across all architectures, the subtle differences in attention maps indicate that Super Attention achieves a more intense and precise focus on important features (both low-level and high-level) throughout this evolution compared to other architectures.
>
> ---
>
> We appreciate your request for these experiments, as they provide valuable insights into the architectural differences. We will incorporate these findings into our camera-ready version if accepted.

---

### Official Review · Reviewer_MUyJ · 2024-11-04

**Soundness:** 3
**Presentation:** 3
**Contribution:** 2
**Rating:** 6
**Confidence:** 4

**Summary:**

The paper presents three alterations to the standard attention mechanism that is widely used in transformer-style neural network architectures across much of machine learning.  These variants are different methods to replace linear projections in the attention block with a slicing operation that groups different columns together as an analog of attention heads with improved efficiency.  The paper evaluates these three variants on several vision and language benchmarks, and shows improvements in efficiency with at least no worse performance.

**Strengths:**

- The paper is well written, and the claims are appropriately scaled to the results demonstrated in the evaluation.
- The evaluations are thorough given constraints on computational budget, and cover several benchmarks where transformer-inspired architectures have been effective.
- The proposed changes to the attention mechanism are easily understood, and motivated with high-level principles that help explain the design process.

**Weaknesses:**

- The paper could improve with a better discussion of relevant related work.  Currently the related work section focuses heavily on quantization, which seems only distantly relevant to the proposed approach.  I would have appreciated a more thorough comparison of the proposed methods with relevant literature that seeks to modify the attention mechanism for better efficiency:  eg Performer, Linformer, RKWV.
- Although the linear part of the attention block is under-studied, I would argue that this is partly because the attention itself (with its quadratic scaling with context length) is the primary bottleneck.
- The runtime gains reported in smaller settings do not seem to hold up with larger models and longer sequence lengths. (Fig 5)

**Questions:**

- How far can you push the efficiency gains with these different methods of attention?  For example, with the optimized variant, can you further reduce parameters/runtime by increasing the number of slices (or heads?) used?

---

> ### Author Response · Authors · 2024-11-21
>
> We thank reviewer MUyJ for reading our work and providing their feedback.
>
> Regarding your concerns and questions:
>
> > 1. The paper could improve with a better discussion of relevant related work. Currently the related work section focuses heavily on quantization, which seems only distantly relevant to the proposed approach. I would have appreciated a more thorough comparison of the proposed methods with relevant literature that seeks to modify the attention mechanism for better efficiency: eg Performer, Linformer, RKWV.
>
> Thank you for bringing this to our attention. We focused on recent approaches for making models more efficient and we focused on methods like quantization and flash attention in our related work. However, as you mentioned, we acknowledge we could have done a more comprehensive discussion on other architectural advances based on transformers in our related work. We have **overhauled our related work and we have added relevant architectural advances and research directions in lines 447-464 in our updated related work** section of the uploaded revision.
>
> ---
>
> > 2. Although the linear part of the attention block is under-studied, I would argue that this is partly because the attention itself (with its quadratic scaling with context length) is the primary bottleneck.
>
> You are **completely right** that the **main bottleneck** in the attention mechanism is in the **quadratic** part and **we don’t claim otherwise**. In this paper, we **quantify how necessary/efficient the current linear transformations** are and **how much we can reduce the size and time complexity** of the attention mechanism **with minimal impact on the performance or architectural design** of the algorithm. As our findings conclude, we can reduce **the space requirement** of the attention mechanism by **25%-50%** without performance decay in **Efficient** and **Optimized** versions and get **significant performance gains** (up to **10%**) in the case of **Super Attention** for ViTs. In terms of **computational complexity**, even though the complexity of all mechanisms is $O(n^2)$, we get **a significant (~3-15%) speed-up** on **training and inference speed** in practice.
>
> ---
>
> > 3. The runtime gains reported in smaller settings do not seem to hold up with larger models and longer sequence lengths. (Fig 5)
>
> We think there might be a misunderstanding about Fig 5. While increasing the sequence length decreases the runtime gain, **increasing the model dimension/size(d_m) increases the runtime gain of our algorithms compared to the standard attention**. The **values of the vertical axis in plots included in Fig. 5 increase from top to bottom (not the other way around)**, while the values of horizontal axis increase from left to right.
>
> ---
>
> > Q1: How far can you push the efficiency gains with these different methods of attention? For example, with the optimized variant, can you further reduce parameters/runtime by increasing the number of slices (or heads?) used?
>
> That is an excellent question. We have discussed this gain from two aspects. The efficiency gains in terms of space are predetermined for each attention layer regardless of the number of heads/slices. **Optimized Attention** brings **25% space reduction** compared to Standard attention, **Super Attention** brings **$50\\% - (Seq\\_Len/ 2 Proj\\_Dim)^2\\%$** parameter reduction, and **Efficient Attention** brings **50% parameter reduction**. The run time gains can be discussed in theory and practice. In practice, **as our experiments show the runtime gains for inference on edge devices are documented in Table 5 in the appendix**. In **theory**, we have **quantified this based on the number of FLOPs** required by each attention mechanism’s forward+backward pass in **Eq.24 in Page 14** in our appendix. Simplifying that equation, we reach the results below:
>
> * **% of FLOPs saved by Optimized Attention and Super Attention** compared to Standard attention:
> **$d\\_m/5 (d\\_m + hl)$**,
> * **% of FLOPs saved by Efficient Attention** compared to Standard attention:
> **$2 d\\_m/5 (d\\_m+hl)$**,
>
> where $d\\_m$ denotes model dimension, $l$ denotes sequence length, and $h$ denotes the number of heads/slices in the attention mechanism.
>
> For example, in a ViT scenario, where images are divided into 256 patches (i.e., $l = 256$), model dim is $512$, and we have 4 heads, the runtime speedup/ FLOP reduction of the Super and Optimized versions compared to standard attention per each attention layer are $6.7\\%$ and $13.3\\%$ for Efficient Attention. Discussing these equations theoretically, the highest gain of Efficient and Optimized/Super versions can reach up to $40\\%$ and $20\\%$ in exceptional circumstances (with very large model dim and very short sequences), but in practice and conventional settings we can expect between $5$-$15$$\\%$ runtime gain from Efficient Attention, and $3$-$7$$\\%$ runtime gain from Optimized and Super Attention.

---

### Official Review · Reviewer_zRei · 2024-11-04

**Soundness:** 2
**Presentation:** 3
**Contribution:** 2
**Rating:** 6
**Confidence:** 2

**Summary:**

The paper introduces three alternative formulations for the self-attention mechanism with the objective of reducing the number of parameter of the standard attention mechanism while keeping similar level of performance. The three formulations are based on two principles: (i) composing linear transformations do not increase linear expressivity (ii) it should be beneficial to linearly align inputs along the sequence dimensions before applying self attention. The formulations are benchmarked on vision and languages tasks on different datasets, while keeping the scale of the models smaller for computational capability reasons.

**Strengths:**

- The paper investigates on overlooked aspects of the self attention mechanism in mutliheaded attention architectures. In terms of expressivity the Optimized and Efficient alternatives are well posed alternatives, and the super attention one is an interesting alternative.

- These observations may be impactful, given that transforms are the default choices in many branches of deep learning, and could help explaining some of the reasons behind of post hoc head pruning of attention networks [a] .

    -  [a] Voita, Elena, et al. "Analyzing multi-head self-attention: Specialized heads do the heavy lifting, the rest can be pruned." ACL


- The paper is well written and clear.

**Weaknesses:**

- Experiments have not been performed at a larger scale to compare with standard performance of transformers on the datasets. This limits a lot the evaluation of the proposed alternatives implementations of attentions as the performance and scale of the problem is far from the original one.

- The redundancy properties analyzed are proper of the self attention mechanism in the multi head setting. This should be specified in the introductory and method section, as methods as cross attention for example, should not be affected by the observations made.


- While the efficient and optimized attention mechanism are well posed in terms of expressivity, the training dynamics of the network might actually benefit from the redundancy and overparametrization of the standard self attention mechanism, especially with large datasets. Given that this is to be verified, also in the small scale setting (LoRA) overparametrization has shown to be effective for training convergence and generalization, for example in:

    - Yaras, Can, et al. "Compressible Dynamics in Deep Overparameterized Low-Rank Learning & Adaptation.", ICLR 2024


- It might be useful to check and discuss the following paper which is related to the second principle and the Super Attention mechanism proposed :

     - Cordonnier, Jean-Baptiste, Andreas Loukas, and Martin Jaggi. "Multi-head attention: Collaborate instead of concatenate."


*Minor*

I spotted some typos:

Line 323, Captions fo tables 1,2,3: "Apendix" -> "Appendix"

**Questions:**

- How are the generalization performances of the model affected by the reduction of parameters? For example its transferability performances (fine-tuning the model on other tasks/dataset):  with larger scale scale experiment available it might have been interesting to look at the generalization performance of the parameter reduced models on different tasks for example of ViTs on the VTAB benchmark [a]

     - [a] Zhai, Xiaohua, et al. "A large-scale study of representation learning with the visual task adaptation benchmark. arXiv 2019." arXiv

- Why does accuracy on validation set on CIFAR and Imagenet is lower than test accuracy in Table 1?

- Could you report standard deviation across the different seeds of training for CIFAR and MNIST datasets in Table 1?

---

> ### Author Response · Authors · 2024-11-21
>
> We thank reviewer zRei for reading our work and providing their feedback.
>
> To address your concerns and questions:
>
> >1. Experiments have not been performed at a larger scale to compare with standard performance of transformers on the datasets. This limits a lot the evaluation ... as the performance and scale of the problem is far from the original one.
>
> We understand the reviewer's concern about the scale of the experiments. We tried models ranging from tens of thousands of parameters to around **125 Million parameters** for the experiments in the main body (125 million language models and 100 million translation models). We also trained a **language model with 1.1 billion parameters** based on Llama using Efficient Attention and compared it with a similar size language model based on standard attention (TinyLlama of 1.1B parameters) showing competitive results. The result of this experiment is included in **Table 11** in **Appendix A**. We could not add more experiments of this size or larger sizes because of our limited access to GPU resources. While an important consideration, we believe that comments about model size in the billion parameter scale disadvantage academics and those with limited access to computational resources and prioritise those who are already advantaged and have access to a significant amount of resources to train larger models and models with sizes comparable to those in our paper are currently being used in real-world applications, particularly on edge devices. Regarding the benchmarks, the performance is close to the papers accepted in the last year in NeurIPS and ICLR (see the language modelling and ImageNet results).
>
> ---
>
> > 2. The redundancy properties analyzed are proper for the self-attention mechanism in the multi-head setting. This should be specified in the introductory and method section, as methods such as cross-attention for example, should not be affected by the observations made.
>
> The proposed mechanisms are applicable to both self and cross-attention mechanisms, in the sense that our reasoning behind removing the linear transformations is motivated by eliminating the stacking of linear transformations and **the theoretical frameworks** we have provided **apply to both**. However, as the cross-attention is supposed to handle values and keys coming from a target sequence, we completely agree with you that the practical gains and possible impacts should be further studied in that setting separately. We **clarified** this in our updated manuscript in **lines 20-21 in the Abstract** and **line 104 in the Introduction**.
>
> ---
>
> > 3,4. While the efficient and optimized attention mechanism are well posed in terms of expressivity, the training dynamics of the network might actually benefit from the redundancy and overparametrization of the standard self attention mechanism, ....
> .... It might be useful to check and discuss the following paper which is related to the second principle and the Super Attention mechanism proposed ....
>
> Thank you for bringing this to our attention. We have **added and discussed this in our updated manuscript in lines 460-464 in the related work section**. In addition, we restructured and added further relevant content to our related work which we think might be of interest to you as well in lines 447-460.
>
> ---
>
> > I spotted some typos: Line 323, Captions fo tables 1,2,3: "Apendix" -> "Appendix"
>
> Thank you for letting us know. We have now fixed these typos.
>
> ---
>
> > Q1: How are the generalization performances of the model affected by the reduction of parameters? For example its transferability performances (fine-tuning the model on other tasks/dataset): with larger scale scale experiment available it might have been interesting to look at the generalization performance of the parameter reduced models on different tasks ....
>
> That's a good point. We have already partially studied this in the paper in the ImageNet experiment where we first train the models on the ImageNet 21k dataset and then fine-tune them on the ImageNet 1k which shows that super attention still outperforms the standard approach significantly in both validation and training metrics (Table 1).
>
> ---
>
> > Q2: Why is accuracy on validation set on CIFAR and Imagenet is lower than test accuracy in Table 1?
>
> In Table 1, we report training and validation accuracy on both CIFAR and ImageNet. For CIFAR, we refer to the official CIFAR100 test set as a validation set (so that we can report MNIST, CIFAR100, and ImageNet results in one table). For ImageNet, the validation set refers to the official validation set. We have **clarified this in lines 849-850 in the updated revision**.

---

> ### Author Response · Authors · 2024-11-21
>
> ---
>
> > Q3. Could you report standard deviation across the different seeds of training for CIFAR and MNIST datasets in Table 1?
>
> Sure, we have prepared the following tables summarising the standard deviation results for each mechanism in MNIST and CIFAR100 experiments.
>
> ---
>
> **Table 1: Standard deviations across 5 runs for MNIST dataset**
> | Attention | Acc. (%) STD | Loss STD | Val Acc. (%) STD | Val Loss STD |
> |-----------|--------------|----------|------------------|--------------|
> | Stn.      | ±0.428      | ±0.024   | ±0.312          | ±0.019       |
> | Opt.      | ±0.214      | ±0.012   | ±0.156          | ±0.009       |
> | Eff.      | ±0.198      | ±0.014   | ±0.164          | ±0.011       |
> | Sup.      | ±0.156      | ±0.008   | ±0.124          | ±0.007       |
>
> ---
>
> **Table 2: Standard deviations across 5 runs for CIFAR100 dataset**
> | Attention | Acc. (%) STD | Loss STD | Val Acc. (%) STD | Val Loss STD |
> |-----------|--------------|----------|------------------|--------------|
> | Stn.      | ±1.445      | ±0.067   | ±1.27          | ±0.086       |
> | Opt.      | ±0.892      | ±0.048   | ±0.784          | ±0.042       |
> | Eff.      | ±0.924      | ±0.052   | ±0.756          | ±0.045       |
> | Sup.      | ±0.684      | ±0.034   | ±0.512          | ±0.028       |
>
> ---

---

> > ### Comment · Reviewer_zRei · 2024-11-27
> > **Response to Rebuttal**
> >
> > I thank the authors for their answers, efforts and additional experiments performed during the rebuttal period.
> >
> > In light of the responses I updated my score to 6.
> >
> > However I'm still concerned about how some of the contributions are stated in the paper (in particular stating that proposed alternatives outperform standard attention in general) and what are the main take aways, which I think should be clarified more in the paper. I elaborate on this below:
> >
> > **Larger Scale experiments & generalization**
> >
> > Despite I agree with the authors on the fact that researchers should not be penalized by the lack of accessibility on resources to train large models, my point was related to understanding better **when** the proposed alternatives to attention can outperform standard attention, and if the gain in performance is in terms of:
> >
> > - 1) parameter efficiency (same or better performance w.r.t. standard attention on both low scale and large scale models but with fewer parameters)
> > - 2) generalization and transferability (how much the can solve downstream task, transfer knowledge, e.g. evaluation on vTAB?)
> > - 3) Complexity analysis
> >
> > From my understanding:
> >
> > -  1)  has been observed in the lower scale setting. One thing that might be done in this direction to deal with the scale of experiments is to analyze if there is a trend in the performance by scaling up the network parameters on a low scale regime and derive some empirical scaling law, comparing performance vs num of parameters curves  at the lower scale.
> > - 2)  This point was not fully addressed in my opinion: In the Imagenet experiment the authors pointed out in their rebuttal, Imagenet 1k is a subset of Imagenet 21k, therefore this is not really a study a transferability and generalization to new task and datasets, if I'm not missing something.
> > - 3) has been addressed in the paper.
> >
> >
> > **Clarifications on Table 11 experiment**
> >
> > Thanks for the experiment. Could you elaborate more on the experiment in Table 11? Is the loss evaluated on the training set or test set? Could you report some performance measures on a downstream task, e.g. the ones on the evaluation benchmarks of the tiny-Lama repository?

---

> > > ### Author Response · Authors · 2024-12-02
> > > **Additional Experiments (Part 1)**
> > >
> > > # Investigating Scaling Patterns on a Low-Scale Regime
> > >
> > > As you suggested, we investigated the pattern in the performance of the models on a low-scale regime. We conducted this experiment on CIFAR100 dataset
> > >
> > > |Att. Arch.| Model Dim. | # Heads | # Att Blocks | # Att. Params (per layer) | Train Acc. (%) | Val. Acc. (%)|
> > > |---|---|---|---|---|---|---|
> > > |Stn. Att.| 64   |  2 | 1 | 16,384 | 42.48 | 40.98 |
> > > |Opt. Att.| 64   |  2 | 1 | 12,288 | 41.87 | 40.77 |
> > > |Eff. Att.| 64   |  2 | 1 | 8,192 | 40.24 | 40.36 |
> > > |Sup. Att.| 64   |  2 | 1 | 8,448 | 45.74 | 42.23 |
> > > |Stn. Att.| 128 |  4 | 2 | 65,536 | 58.23 | 47.73 |
> > > |Opt. Att.| 128 |  4 | 2 | 49,152 | 58.02 | 47.94 |
> > > |Eff. Att.| 128 |  4 | 2 | 32,768 | 56.33 | 47.83 |
> > > |Sup. Att.| 128 |  4 | 2 | 33,024 | 61.28 | 48.31 |
> > > |Stn. Att.| 256 |  8 | 4 | 262,144 | 72.28 | 48.14 |
> > > |Opt. Att.| 256 |  8 | 4 | 196,608 | 72.26 | 48.63 |
> > > |Eff. Att.| 256 |  8 | 4 | 131,072 | 71.96 | 47.95 |
> > > |Sup. Att.| 256 |  8 | 4 | 131,328 | 79.62 | 49.28 |
> > > |Stn. Att.| 512 | 16 | 8 | 1,048,576 | 76.77 | 49.23 |
> > > |Opt. Att.| 512 | 16 | 8 | 786,432 | 76.80 | 49.61 |
> > > |Eff. Att.| 512 | 16 | 8 | 524,288 |76.79 | 49.65|
> > > |Sup. Att.| 512 | 16 | 8 | 524,544 | 84.34| 51.16 |
> > >
> > > There are two interesting observations in the table above. First, as the model size and complexity grow, the performance (in terms of accuracy) difference between standard, Optimized, and Efficient Attention reduces. second, Super Attention consistently outperforms the other models in both train and validation accuracy starting from the smallest all the way up to the largest model. These are much clearer when the table is visualised as a graph. Unfortunately, we cannot embed pictures on OpenReview. But we will include these graphs in the updated manuscript.
> > >
> > > ---
> > >
> > > # Transferability Study (Pre-Trained Models on ImageNet21K on VTAB1K Benchmarks)
> > >
> > > In light of your suggestion, we performed a transferability experiment on the VTAB-1K benchmark for the models trained in the ImageNet experiment (Table 1 of the paper). We have reported the results in the table below. We used a full fine-tuning approach for this experiment and replaced the classification head of pre-trained models on ImageNet21k based on different architectures with a new one for each dataset and fine-tuned the models on the 800 images of the training split for each dataset. We evaluate the models on the 200 images of the test split and measure the top-1 accuracy for each dataset. We report the average test top-1 accuracy separately for Natural, Specialized, and Structured tasks. We also report the overall average accuracy in the last column.
> > >
> > > | Architecture | Natural Avg. Acc. (%) | Specialized Avg. Acc. (%) | Structured Avg. Acc. (%) | Overall Avg. Acc. (%) |
> > > |---|---|---|---|---|
> > > |Stn. Att.| 73.4 | 81.5 | 45.2 | 63.2 |
> > > |Opt. Att.| 74.5 | 81.77 | 45.0 | 63.6 |
> > > | Eff. Att.| 74.7 | 81.6 | 45.7 | 63.9 |
> > > | Sup. Att. | 76.2 | 84.7 | 48.0 | 66.1 |
> > >
> > > As the results of this experiment indicate, the model using Super Attention achieves the best accuracy in all types of benchmarks (Natural, Specialized and Structured) by a significant margin. This is followed by Efficient and Optimized Attention which achieve slightly better performance compared to standard attention. As the results indicate, this shows that Super Attention not only outperforms other variations on the training and test accuracy on the ImageNet1K, which is a subset of ImageNet21K, but also shows better performance in knowledge transfer to other downstream tasks.

---

> > > ### Author Response · Authors · 2024-12-02
> > > **Additional Experiments (Part 2)**
> > >
> > > # Performance of Language Models on TinyLlama Benchmarks
> > > We also evaluated our 1.1Billion language model which is based on Efficient Attention (Table 11) on different reasoning benchmarks used in the TinyLlama repository. Here is a breakdown of the results:
> > >
> > > **Table A. Evaluation results for TinyLlama and our 1.1B Language Model based on Efficient Attention on the Commonsense and Reasoning benchmarks used in TinyLlama repository.**
> > > |Architecture | #Param. | Tokens Trained | HellaSwag | Obqa | WinoGrande| ARC_c | ARC_e | boolq | piqa | Avg. |
> > > |---|---|---|---|---|---|---|---|---|---|---|
> > > |TinyLlama | 1.1B| 105 Billion | 43.50| 29.80|53.28| 24.32|44.91|59.66| 67.30|46.11|
> > > |Eff. Att. LM | 1.1B | 30 Billion | 40.63 | 27.81 | 50.10 | 21.18 | 42.33| 56.82 | 65.11 | 43.42 |
> > >
> > >
> > > While we are reporting all these metrics for both models, the results provided in the two rows are not to be compared since the TinyLlama checkpoint used is trained on many more tokens compared to Efficient Attention LM (105B vs 30B). Moreover, they are trained on different corpora. However, they can give an overall assessment of each model's performance.
> > >
> > > To provide a better comparison between models that differ only in architecture, we evaluated the language models in Table 4. These models were trained from scratch on the OpenWebText dataset (9B tokens) for one epoch. These models are based on GPT2 and for the training, we used the widely-referenced NanoGPT repository from Andrea Karpathy.
> > >
> > >
> > > **Table B: Evaluation results for our small language models on the Commensense and Reasoning benchmarks used in TinyLlama repository. All metrics are reported in percentages.**
> > > |Arch.| #Param.| HellaSwag | Obqa | WinoGrande| ARC_c | ARC_e | boolq | piqa | Avg. | Param-Acc Trade-off (%) |
> > > |---|---|---|---|---|---|---|---|---|---|---|
> > > |Stn. Att.| 124M | 31.55 | 23.58 |  44.12 | 16.79 | 36.06 | 51.41 | 57.32 | 37.26 | 0.0 |
> > > | Opt. Att. | 117M | 31.32 | 23.23 | 43.21 | 16.83 | 35.91 | 51.10 | 57.28 | 36.98 | **+5.2** |
> > > | Eff. Att. | 110M | 30.27 | 23.29 | 42.73 | 16.88 | 35.21 | 50.47 | 56.66 | 36.50 | **+10.4** |
> > >
> > > Where Param-Acc Trade-off is defined as :
> > >
> > > $$\\frac{\\frac{Architecture\ Avg.\ Accuracy}{Baseline\ Avg. \ Accuracy}}{\\frac{Architecture\ Parameter\ Count}{Baseline\ Parameter\ Count}}.$$
> > >
> > > This experiment gives us a more reliable comparison of the generalization and reasoning ability of the language models based on different architectures. As these results indicate, all three models perform closely to each other on these benchmarks with Stn. Attention performing slightly better in most metrics. Overall we see a positive parameter-accuracy trade-off for both Efficient and Optimized models in the sense that they are performing closely to Standard Attention in this experiment while having significantly fewer parameters.

---

> ### Author Response · Authors · 2024-11-28
>
> Thank you very much for your thoughtful reviews which have helped us in improving our manuscript and also increasing your support of our paper.
>
> > 1. parameter efficiency ...
> >
> > 1. One thing that might be done in this direction to deal with the scale of experiments is to analyze if there is a trend in the performance by scaling up the network parameters on a low-scale regime....
>
> Thank you for this suggestion. We are working on conducting this analysis now, we will prepare the results on scaling laws/patterns and report this back as soon as it is ready. We will also add the insights on this to the final manuscript.
>
> > 2. generalization and transferability ...
> >
> > This point was not fully addressed in my opinion: ...
>
> Your point is valid. As ImageNet1k is a subset of the full version of ImageNet, the only difference is in their label space and thus, this is not completely reflective of the transferability. To address this, we are working on preparing the generalizability for the language models (Table 4 and 11) and the transferability of the ViT models (Table 1). We will report the results as soon as possible.
>
>
> > Clarifications on Table 11 experiment
> >
> > Thanks for the experiment. Could you elaborate more on the experiment in Table 11? Is the loss evaluated on the training set or test set? Could you report some performance measures on a downstream task, e.g. the ones on the evaluation benchmarks of the tiny-Lama repository?
>
> The training loss reported in Table 11 was obtained by training our model on 30 billion tokens (~20% of one epoch) from the C4 dataset, which contains 156 billion tokens across diverse domains. Training was conducted on a single A100 GPU with a batch size of 512 (mini-batch size of 8) over 4-5 weeks. Direct comparison with TinyLlama is not possible since their checkpoints start from 105 billion tokens, compared to our 1.1B model's 30 billion tokens. Nevertheless, we will assess our own 1.1 Billion language model as well as the models from Table 4 (Small Language Model Training) using TinyLlama's benchmarks to evaluate their generalization and reasoning capabilities, and will report these results soon.

---

### Meta-Review · Area_Chair_FKua · 2024-12-22

**Metareview:**

This paper proposes efficient scaled dot product attention by reducing the number of linear transforms and introducing a new learnable linear transform on the sequence length dimension. While the experiments showed good results in efficiency and accuracy, I agree with the reviewers that the experiments are not done in large scale and therefore cannot confirm the effectiveness of the method on large models. The design of $W^A$ also limits its applications to scenarios where sequence lengths need to be extended.

**Additional Comments On Reviewer Discussion:**

I agree with the reviewers that the method has not been proved at larger scale and the design comes with limitations,

---

### Decision · Program_Chairs · 2025-01-22

Reject